# Interplay between α2-chimaerin and Rac1 activity determines dynamic maintenance of long-term memory

Li Lv[1,4], Yunlong Liu[1,4], Jianxin Xie[2,3,4], Yan Wu[1], Jianjian Zhao[1], Qian Li[1] & Yi Zhong[1]*

Memory consolidation theory suggests that once memory formation has been completed, memory is maintained at a stable strength and is incapable of further enhancement. However, the current study reveals that even long after formation, contextual fear memory could be further enhanced. Such unexpected enhancement is possible because memory is dynamically maintained at an intermediate level that allows for bidirectional regulation. Here we find that both Rac1 activation and expression of α2-chimaerin are stimulated by single-trial contextual fear conditioning. Such sustained Rac1 activity mediates reversible forgetting, and α2-chimaerin acts as a memory molecule that reverses forgetting to sustain memory through inhibition of Rac1 activity during the maintenance stage. Therefore, the balance between activated Rac1 and expressed α2-chimaerin defines dynamic long-term memory maintenance. Our findings demonstrate that consolidated memory maintains capacity for bidirectional regulation.

[1] Peking University-Tsinghua University-National Institute Biological Science Joint Graduate Program, Tsinghua-Peking Center for Life Sciences, School of Life Sciences, Tsinghua University, Beijing 100084, China. [2] National Laboratory of Biomacromolecules, Institute of Biophysics, Chinese Academy of Sciences, 15 Datun Road, Chaoyang District, Beijing 100101, China. [3] University of Chinese Academy of Sciences, Beijing 100101, China. [4] These authors contributed equally: Li Lv, Yunlong Liu, Jianxin Xie *email: zhongyithu@tsinghua.edu.cn

Memory consolidation theory suggests that a newly acquired labile memory is gradually consolidated or stabilized through gene-regulation-based persistent modifications in synaptic structures within neural circuits and the resulting consolidated long-term memory (LTM) is maintained at a stable strength[1–3]. Guided by such widely accepted theory, the study of memory maintenance is understandably devoted to the identification of learning-stimulated self-sustainable protein synthesis regulatory mechanisms[4,5]. Indeed, a number of candidate mechanisms for such a simple maintenance have been reported, including the autophosphorylation of calcium/calmodulin-dependent protein-kinase type II (CAMKII)[6,7], and the persistent protein synthesis regulated via self-perpetuating prion-like translational factors such as the cytoplasmic polyadenylation binding protein-3 (CPEB-3)[8]. It is a common experience, however, when recalling an episode that happened some time ago, the memory is vivid at one time but vague at others. Such flexible recalling experiences seem to be counterintuitive to memory consolidation theory. In addition to common experiences, it is noteworthy that recent publications have reported a number of intriguing observations that appear inconsistent with the notion of stabilized memory. First, synaptic connections involved in maintaining LTM are highly dynamic[5]. Second, overexpression of the self-autonomous protein-kinase-M-zeta (PKMζ)[9] was reported to enhance memory long after memory formation, but mechanisms underlying such enhancement remain unexplained[10].

Forgetting, the flip side of memory consolidation, is also an important feature for different types of memories[11]. The recently proposed concept of active forgetting describes an intrinsic mechanism, i.e., learning or training evokes active forgetting to accelerate the decay of a formed memory. Multiple mechanisms are attributed to forgetting, including through activation of Rac1 or Cdc42, as well as through neurogenesis[11]. Recent progress in study of forgetting reveals that suppression of forgetting, inhibition of Rac1-dependent forgetting in particular, is capable of prolonging an hour-long labile memory to days[12–14] in a range of tasks from invertebrates to vertebrates[15,16]. Moreover, a study reports that LTM decayed away during infancy can be recovered in adulthood[17], suggesting that neurogenesis-based forgetting does not lead to memory erasure. These observations got us interested in investigating whether forgetting mechanisms, Rac1-dependent forgetting in particular, play any roles in memory maintenance. One possible candidate molecule to inhibit Rac1 activity comes from recent studies in mice during developmental stages[18–21]. It reveals that the negative regulator of Rac1 activity, α2-chimaerin, acts as a brake through its GTPase-activating function to constrain Rac1 activity in regulating hippocampal dendrite and spine morphogenesis[20,22] for establishment of normal cognitive ability[23]. However, the roles of interaction between Rac1 activity and α2-chimaerin in memory maintenance in adulthood remain to be unraveled.

The present study provides a novel mechanism for dynamic memory maintenance that affords an intuitive explanation as to why LTM is still capable of being enhanced or faded long after completing formation. By integrating genetic, pharmacological, and optogenetic manipulations during behavioral paradigms, we demonstrate that the interplay between learning-evoked Rac1-dependent forgetting and learning-induced expression of a memory molecule α2-chimaerin enables consolidated contextual fear memory and is maintained at an intermediate level so that memory could be enhanced through expression of more α2-chimaerin while attenuated through elevated Rac1 activity. Moreover, forgetting induced by enhanced Rac1 activity is reversible through suppression of Rac1 activity. Thus, our results reveal that consolidated memory is maintained through the learning-evoked Rac1-dependent reversible forgetting mechanism balanced by learning-evoked expression of a memory maintenance molecule, α2-chimaerin.

## Results

**Inhibition of Rac1 activity causes memory enhancement.** As the suppression of active forgetting prolongs hour-long short-term memory to days[12,14], we were interested in studying whether Rac1-dependent forgetting mechanisms play roles in LTM maintenance. In an effort to investigate this issue, we assayed Rac1 activity in hippocampal extracts of mice trained in the contextual fear conditioning (CFC) during the maintenance period of memory. Wild-type (WT) mice were subjected to single- or three-trial CFC and were tested for memory retention at different time points (1 h, day 1, day 7, day 14, or day 36) (Fig. 1a and Supplementary Fig. 1a). To avoid the confounding effects of extinction, separate groups of mice were tested at each time point. Consistent with previous reports[7], mice exposed to three-trial CFC retained a much higher and stable contextual fear memory across the entire testing period (Supplementary Fig. 1a), whereas single-trial CFC produced a protein synthesis-dependent LTM[8], which was maintained stably for at least 7 days (Fig. 1a) and then decayed from day 7 to day 14, and reached a stable bottom level (dashed line) until day 36 (Fig. 1a). Considering that dynamic memory maintenance, the current work was mainly devoted to single-trial CFC-induced contextual fear memory. Immunoblotting showed that single-trial CFC did not activate Rac1 initially at 1 h after training, but significantly increased Rac1 activity from day 1 to days 2, 7, and 11, and then the activated Rac1 returned to the basal level on day 14 (Fig. 1b). Consistently, immunofluorescence staining showed about 50% of hippocampal CA1 cells were labeled with the evoked Rac1 activation (Fig. 1c and Supplementary Fig. 1b). In addition, neither context nor footshock alone activated Rac1 (Supplementary Fig. 2a). However, three-trial CFC caused the decreased Rac1 activity (Supplementary Fig. 2d).

The observation in single-trial CFC prompted us to determine whether training-stimulated Rac1 activity mediates active forgetting of contextual fear memory, as in the case of novel object recognition memory[15]. Mice were bilaterally injected with adeno-associated viruses (AAVs), carrying mutant transgenes that encoded either dominant-negative Rac1 (Rac1-DN) to inhibit endogenous Rac1 activity or constitutively active Rac1 (Rac1-CA) to increase Rac1 activity, targeted to hippocampal excitatory neurons[15] (Supplementary Fig. 3a). Expression of enhanced green fluorescent protein (EGFP) and corresponding changes in Rac1 activity were confirmed (Supplementary Fig. 3b, c). Moreover, such constitutive activation or inhibition of Rac1 did not cause permanent nonspecific hippocampal damage (Supplementary Fig. 3d). We noted that 1 h memory after training was similar (Fig. 1d), regardless of whether Rac1 activity was enhanced or inhibited, which is consistent with the notion that Rac1 activity has no impact on the memory formation[12,15,16]. However, elevated Rac1 activity accelerated 24 h memory decay (the bottom level in Fig. 1d), whereas reduced Rac1 activity led to memory enhancement (Fig. 1d), irrespective of memory formation. To corroborate this observation, we also assayed the effects of post-training inhibition of Rac1 activity through injection of the Rac1 inhibitor Ehop016[15,24,25] immediately after training. Reduction of Rac1 activity via Ehop016 injection enhanced memory at 24 h, without affecting memory at 1 h (Supplementary Fig. 4a, b). Thus, CFC-stimulated Rac1 activity should accelerate memory decay. Conversely, Rac1 activity inhibition causes memory enhancement.

As long-lasting activation of Rac1 following learning is observed in the hippocampus (Fig. 1b, c and Supplementary

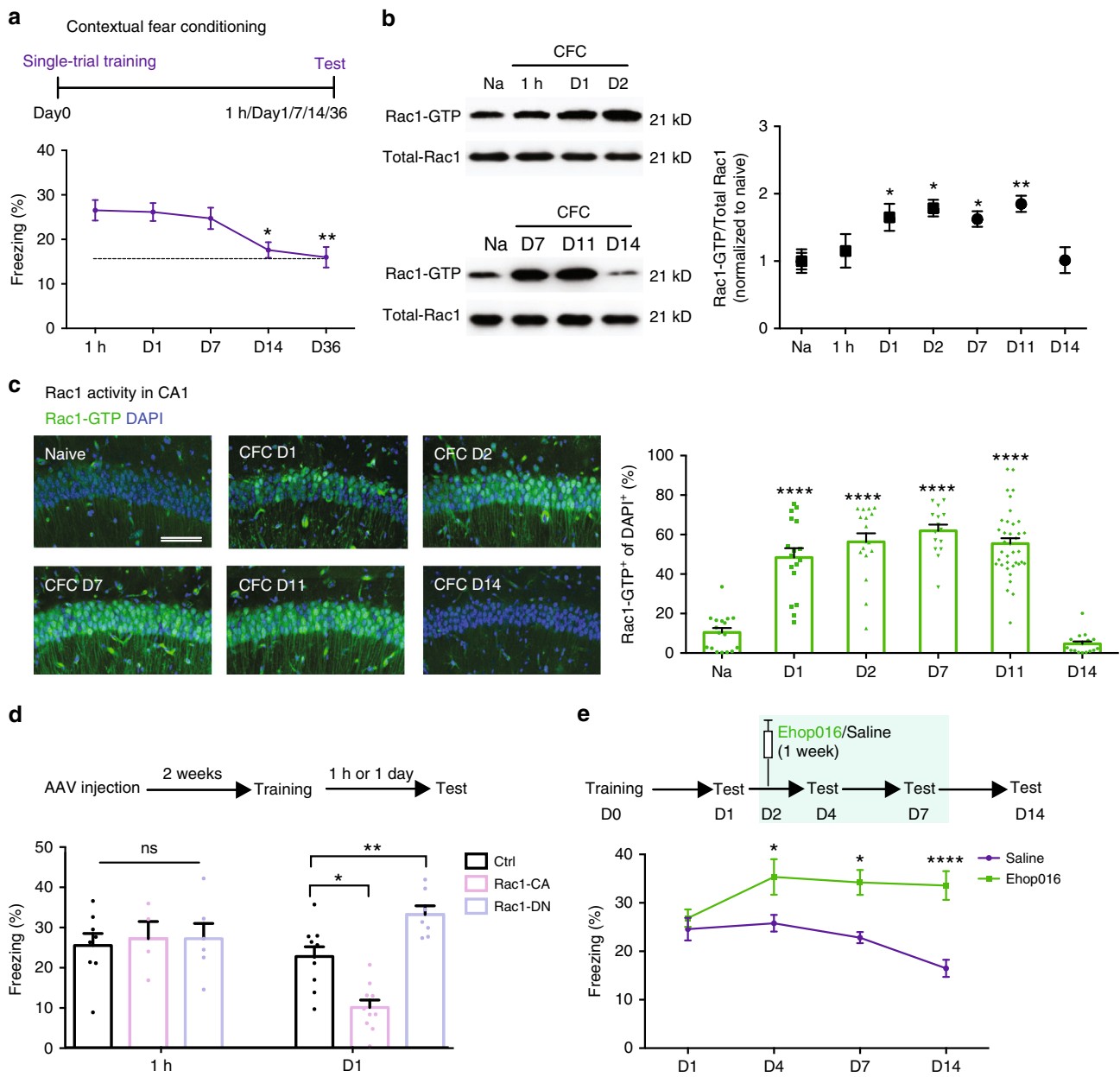

**Fig. 1** Inhibition of learning-induced Rac1 activity causes memory enhancement. **a** Behavioral schedule (top) for contextual fear conditioning (CFC)-induced memory retention curve (D, day) in wild-type (WT) mice (bottom). WT mice were trained using one-shock CFC and then separate group mice were tested for memory retention at various periods (1 h, day 1, day 7, day 14, or day 36). *$P < 0.05$ and **$P < 0.01$ (from one-way ANOVA); $n = 41, 28, 17, 22$, and 21 mice. **b** Representative image of western blottings (left) and data (right) showing total Rac1 levels and Rac1-GTP in hippocampal extracts from naive (Na) and trained (one shock) at various retention intervals (1 h, day 1, day 2, day 7, day 11, or day 14). *$P < 0.05$ and **$P < 0.01$ (from one-way ANOVA); $n = 12$, 7, 7, 3, 4, 4, and 4 mice. **c** Representative immunostaining of Rac1-GTP (left) and data (right) showing percentages of Rac1-GTP+ cells in the hippocampal CA1 region of Na mice and CFC mice at various periods (day 1, day 2, day 7, day 11, or day 14). Coronal sections of the hippocampal CA1 region of Na mice and CFC mice with the anti-Rac1-GTP (green) and anti-nuclei (blue). Scale bar, 500 μm. ****$P < 0.0001$ (from one-way ANOVA); $n = 15$-17 slices from 7 mice for each group. **d** Behavioral effects of the bidirectional manipulation of hippocampal Rac1 activity on contextual fear memory. After AAV injection for 2 weeks, mice were subjected to one-shock CFC and then separate group mice were tested for 4 min at different time periods (1 h or day 1). *$P < 0.05$ and **$P < 0.01$ (from two-way ANOVA); $n = 8, 10, 6, 7, 4$, and 10 mice. **e** Time courses of freezing behavior (bottom) in WT mice with injection of Ehop016 or saline (top). *$P < 0.05$ and ****$P < 0.0001$ (from two-way ANOVA); $n = 10$, 14 mice. Data are presented as means ± SEM. Also see Supplementary Table 1.

Fig. 1b), we hypothesized that memory might be enhanced through the suppression of Rac1 activity during the maintenance period, even long after memory formation. To further test this hypothesis, we assayed the effects of Rac1 activity inhibition following the injection of Ehop016 during the maintenance stage. Ehop016 was administered from day 2 after CFC, once per day,

for 1 week (Fig. 1e, top), which presumably long after the memory had formed. In this case, multiple injection of Ehop016 did not cause permanent nonspecific hippocampal damage (Supplementary Fig. 4c). In addition, such pharmacological reduction of hippocampal Rac1 activity had no effects on locomotion or anxiety (Supplementary Fig. 4d, e). Indeed, these

mice showed normal memory at day 1. Intriguingly, Rac1 pharmacological inhibition enhanced memory in Ehop016-treated mice at days 4, 7, and 14, in comparison with memory in saline-treated mice (Fig. 1e). Moreover, this memory enhancement was retained far beyond the maintenance period and required the continued suppression of Rac1 activity (Fig. 1e). Normally, the memory decays to a stable bottom level at day 14 (Fig. 1a). Together, inhibition of learning-stimulated hippocampal Rac1 activity is required for the enhancement of consolidated memory.

**Rac1 activity within engram cells affects memory maintenance.** Hippocampal CA1 engram cells critically contribute to the maintenance of contextual fear memory[26,27]. We noticed that dorsal hippocampal Rac1 activity was robustly elevated by CFC (Fig. 1b, c and Supplementary Fig. 1b). We thus asked: could this long-lasting activated Rac1 activity act as a memory suppressor, but not as an eraser within the CA1 engram cells for dynamic memory maintenance? This query leads us to a hypothesis of reversible forgetting-based dynamic memory maintenance that can be described as follows: the memory from day 1 to day 7 is normally maintained at intermediate levels so that bidirectional regulation is permitted.

The following three experiments support this hypothesis. First, we tested whether activation of Rac1 following learning allocates within the CA1 engram cells. To visualize CFC-activated Rac1 activity within engram cells, we bilaterally injected AAV$_9$-c-fos: tetracycline-controlled transactivator (tTA) and AAV$_9$-tetracycline response element (TRE)-Tandem dimer Tomato (tdTomato) to the CA1 region of WT mice (Fig. 2a). Moreover, we examined the viral expression under various treatments, including CFC, context-only, and shock-only (Fig. 2b, c). Upon doxycycline (Dox) withdrawal, engram cells activated by CFC, context-only, or shock-only were labeled with tdTomato[28,29] (Fig. 2b, c), whereas activated Rac1 under such various treatments in CA1 neurons were labeled with a Rac1-GTP antibody (Fig. 2c and Supplementary Fig. 5a). We observed that CFC, context alone, or shock alone was sufficient to induce tdTomato expression under 2 days off Dox (Fig. 2c and Supplementary Fig. 5a). In particular, the number of c-fos$^+$ cells increased substantially in response to CFC (Fig. 2c, d). Next, we calculated the proportion of double-labeled cells (Fig. 2c and Supplementary Fig. 5a). Immunohistochemistry against activated Rac1 (Rac1-GTP$^+$) was assessed by the percentage of Rac1-GTP$^+$/c-fos$^+$ colabeled cells among c-fos$^+$ cells (engram cells). We found that 79% of c-fos$^+$ cells were labeled with activated Rac1 positive (Rac1-GTP$^+$) at 24 h following training (Fig. 2e). The percentages of double-labeled cells in the group with context alone or shock alone were low (7.4% and 4.4%, respectively; Fig. 2e). Moreover, 10% of non-engram cells were labeled with active Rac1 positive (Supplementary Fig. 5b), demonstrating that learning-specific activated Rac1 allocates within the CA1 engram cells.

Second, we investigated whether memory could be suppressed at any time during the maintenance period through additional activation of Rac1 within engram cells and such forgotten memory could be salvaged through Rac1 activity inhibition. The experimental (PaRac1) mice were injected bilaterally into the CA1 with a virus cocktail of AAV$_9$-c-fos:tTA and AAV$_9$-TRE:PaRac1-tdTomato to express photoactivatable Rac1 (PaRac1) in CA1 engram cells and the control (tdTomato) mice were injected with AAV$_9$-c-fos:tTA and AAV$_9$-TRE-tdTomato (Fig. 2f, top). In the absence of Dox, training-induced neuronal activity selectively labels active c-fos-expressing CA1 neurons with PaRac1-tdTomato, thereby allowing for the optogenetic activation of Rac1 at any given time in CA1 engram cells[30–32]. We confirmed that CA1

engram cells were labeled with PaRac1-tdTomato (Fig. 2f, bottom). Activation of Rac1 in CA1 engram cells by blue light (Fig. 2g and Supplementary Fig. 6a) at day 4 indeed caused significant and immediate memory decay. However, memory recovered to a normal or intermediate level at day 7 (Fig. 2g). Next, we tested light stimulation-induced PaRac1 activation by measuring the level of Rac1 effector phospho-PAK[31] in PaRac1-expressing engram cells. Indeed, light stimulation led to increased phospho-PAK in PaRac1-expressing engram cells at day 4 and then such increased phospho-PAK returned to the basal level at day 7, in comparison with mice that were not subjected to light (Supplementary Fig. 6b-d). Furthermore, optogenetically induced memory decay could also be rapidly restored to an even higher level, through the injection of Ehop016 after light stimulation (Fig. 2h), accompanied with the return of increased p-PAK to the basal level (Supplementary Fig. 6b-d). To further confirm that blue-light-induced memory decay indeed resulted from increased Rac1 activity, Ehop016 was injected prior to light stimulation (Fig. 2i, top). In this case, blue-light exposure was unable to cause memory decay (Fig. 2i, bottom). Moreover, optogenetic stimulation had no impact on locomotion or anxiety (Supplementary Fig. 6e), which are both critical behavioral parameters for the CFC task.

The findings from the above two experiments led us to make a third prediction that the naturally faded memory for several days (staying on day 14; see also Fig. 1a) could be enhanced through the suppression of Rac1 activity. Injection of the Rac1 inhibitor for seven consecutive days beginning at day 14 led to recovered memory on day 22 (Fig. 2j). Thus, activation of Rac1 within engram cells in the CA1 region is sufficient to cause memory decay at any time during the maintenance period. Such optogenetically induced forgotten memory can be rapidly salvaged through Rac1 pharmacological inhibition, which also recovers naturally faded memory, even several days after the memory had decayed away.

**α2-Chimaerin increases following learning are long lasting.** If active forgetting is launched following learning, then memories should fade fast[12,15,16]. An obvious question is raised as to how contextual fear memory could be maintained at an intermediate level when facing such sustained Rac1-dependent active forgetting (Fig. 1). Subsequent efforts led to the discovery of CFC-stimulated expression of a Rac-specific GTPase-activating protein (Rac-GAP), α2-chimaerin, which is known to inactivate Rac1 activity in the nervous system[18,19,21]. Single-trial CFC induced increased expression of α2-chimaerin from day 1 to day 7, but not at 1 h, and this expression returned to the basal level on day 11 (Fig. 3a).

To verify the protein synthesis of α2-chimaerin after single-trial CFC, we utilized a protein synthesis inhibitor anisomycin (ANI). Additional injections of ANI further supported the finding (Fig. 3b). This expression is specific to single-trial CFC but not context alone, shock alone, and three-trial CFC (Supplementary Fig. 2b, e). Furthermore, the expression of its homolog α1-chimaerin was not stimulated by the CFC (Supplementary Fig. 2c).

**α2-Chimaerin interacts with Rac1 to maintain memory.** To determine α2-chimaerin functional roles, we manipulated expression levels of α2-chimaerin in the hippocampal CA1 region in which its highest endogenous expression is reported[23]. For knockdown, we generated AAVs that carry a short hairpin RNA (shRNA) sequence[22] that targets α2-chimaerin (AAV-α2 shRNA), fused to tdTomato, under the control of the cytomegalovirus (CMV) early enhancer/chicken β-actin (CAG)

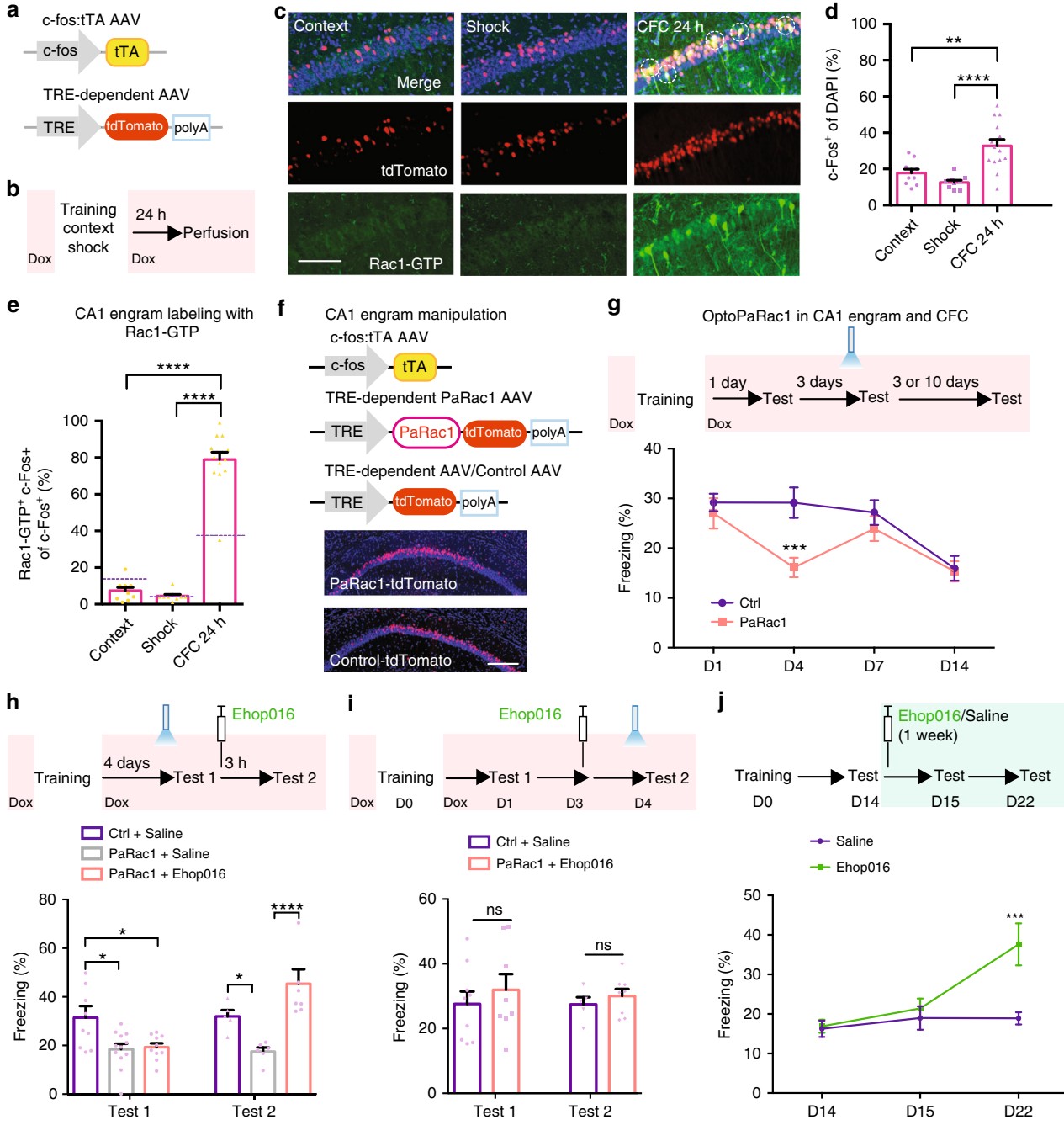

**Fig. 2** Regulation of Rac1 activity within CA1 engram cells affects memory maintenance. **a–d** Strategy of CFC-activated CA1 engram cells labeling with Rac1 activity immunoreactivity. **a** Diagram of AAV$_9$-c-fos:tTA or AAV$_9$-TRE-tdTomato. **b** Experimental schedule. **c** Coronal section of CA1 engram cells (red) labeling with anti-Rac1-GTP (green). Scale bar, 100 μm. Circled cells represent examples of double-positive. **d** Percentages of engram cells in CA1. **\*\*$P <$ 0.01 and \*\*\*\*$P <$ 0.0001 (from one-way ANOVA); $n =$ 10, 9, 14 mice. **e** Percentages of CA1 engram cells labeling with anti-Rac1-GTP. Chance levels, indicated by grape dashed lines, were estimated at 14.2% (context), 5.1% (shock), and 38.2% (CFC 24 h). \*\*\*\*$P <$ 0.0001 (from one-way ANOVA); $n =$ 10, 9, 14 mice. **f** CA1 engram cells express PaRac1. Diagram of AAV$_9$-c-fos:tTA, AAV$_9$-TRE:PaRac1-tdTomato, and AAV$_9$-TRE-tdTomato (top), and coronal section of the hippocampal CA1 region with PaRac1-tdTomato (red) (bottom). Scale bar, 300 μm. **g–i** Experimental schedule (top) and time courses of freezing behavior (bottom). \*\*\*$P <$ 0.001 (from two-way ANOVA); $n =$ 11, 17, 15, 11, 6, 12, 13, 13 mice (**g**). **h** Test 1 was performed 1.5 h after light stimulation at day 4. Then, Ehop016 or saline was injected into the mice immediately after Test 1. Three hours after Ehop016 or saline injection, Test 2 was performed. \*$P <$ 0.05 and \*\*\*\*$P <$ 0.0001 (from two-way ANOVA); $n =$ 9, 5, 10, 7, 12, 6 mice. **i** One day after CFC, Test 1 was performed. Two days after Test 1, mice were injected with Ehop016 or saline. One day later, mice received light stimulation and then Test 2 was performed 1.5 h after the light stimulation. $n =$ 9, 5, 8, 8 mice (two-way ANOVA). **j** Time courses of freezing behavior (bottom) in WT mice with injection of Ehop016 or saline once per day for 1 week (top). \*\*\*$P <$ 0.001 (from two-way ANOVA); $n =$ 4, 4, 4, 15, 11, 7 mice. Data are presented as means ± SEM. Light red shading signifies that mice are on DOX (**b**, **g**, **h**, **i**). Data are presented as means ± SEM. Also see Supplementary Table 1.

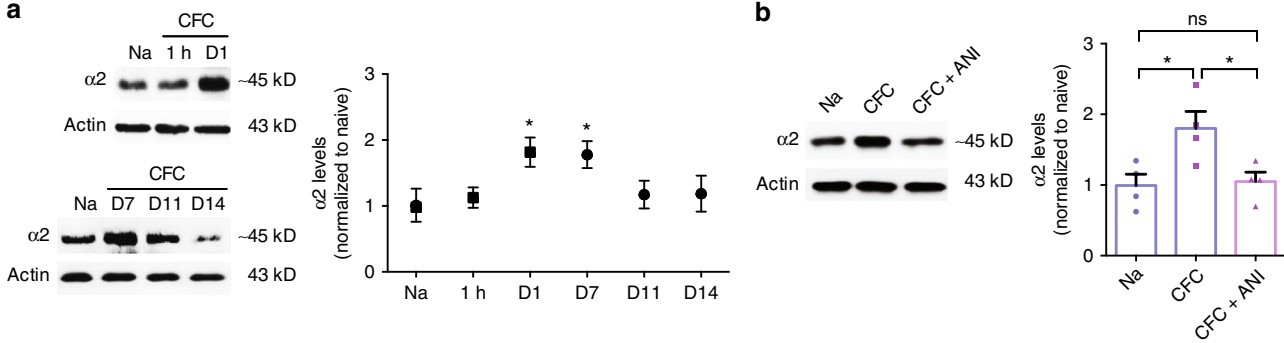

**Fig. 3** Learning-evoked the synthesis of Rac-GAP α2-chimaerin. **a** Representative image of western blottings (left) and data (right) showing hippocampal α2-chimaerin of Na mice and CFC mice. *$P < 0.05$ (from one-way ANOVA); $n = 12, 4, 6, 5, 6, 5$ mice. **b** Immunoblotting exhibiting hippocampal α2-chimaerin of Na mice, CFC mice, and CFC mice with ANI treatment (CFC+ANI) mice at 1 day after training. *$P < 0.05$ (from one-way ANOVA); $n = 4, 4, 4$ mice. Data are presented as means ± SEM. Also see Supplementary Table 1.

promoter (Fig. 4a, left). For overexpression, we constructed AAVs that carry an α2-chimaerin encoding gene (AAV-α2-chimaerin), fused to EGFP under the control of the calcium/calmodulin-dependent protein-kinase IIα (CaMKIIα) promoter (Fig. 4a, middle). Subsequently, mice were injected with either AAV-α2 shRNA or AAV-α2-chimaerin, as well as their controls (AAV-scramble shRNA or AAV-control, respectively), in the dorsal hippocampus (Fig. 4a, right). The α2-chimaerin knockdown and overexpression effects were confirmed (Fig. 4b–f), along with the elevated and reduced hippocampal Rac1 activity, respectively (Fig. 4g, h). In addition, knockdown of α2-chimaerin (α2 shRNA) enhances the tendency of Rac1 activity increase following CFC compared with trained (CFC) WT mice (Fig. 4i), validating that partial inhibition of learning-induced Rac1 activity is indeed a result of α2-chimaerin synthesis. We also found that manipulation of Rac1 activity had no feedback effect on α2-chimaerin expression (Supplementary Fig. 3e). Moreover, the viral manipulation of α2-chimaerin expression had no impact on locomotion and anxiety (Supplementary Fig. 7a, b).

Knockdown of α2-chimaerin caused rapid memory decay without affecting memory at 1 h. Memory at 24 h was reduced to a bottom level similar to that in controls at day 14 and remained at that level (Fig. 4j). Conversely, α2-chimaerin overexpression enhanced 24 h memory to a significantly higher level than that in the ctrl shRNA (ctrl sh) group, which was sustained until day 14 (Fig. 4k). Such a bidirectional memory regulation is resulted from α2-chimaerin-regulated Rac1 activity. Indeed, knockdown of α2-chimaerin underlying pharmacological inhibition of Rac1 activity via Ehop016 (Supplementary Fig. 4f) not only failed to reduce memory but, in fact, also greatly enhanced memory from day 1 through day 14 (Fig. 4l). These data, combined with the results in Fig. 1, substantiate that the suppression of Rac1 activity by the α2-chimaerin synthesis following learning enables the maintenance of an intermediate-level memory state, which allows for dynamic memory maintenance when α2-chimaerin expression is altered.

We also studied the effects of manipulating another member of the α-chimaerin family, α1-chimaerin[21]. We found that manipulation of α1-chimaerin expression had no effects on either the memory maintenance induced by single-trial CFC (Supplementary Fig. 8a-e) or on the relevant behavioral parameters (locomotion and anxiety) (Supplementary Fig. 8f). Furthermore, double knockdown of α1-chimaerin and α2-chimaerin had no effects on the relevant behavioral parameters, but yielded an accelerated memory decay phenotype (Supplementary Fig. 9) that is similar to the effects of knockdown of only α2-chimaerin

(Fig. 4j). These data further support a key role of α2-chimaerin, but not α1-chimaerin, in memory maintenance.

**Knockdown of α2-chimaerin impairs the multiple types of memories**. To examine whether the roles of α2-chimaerin in memory maintenance induced by single-trial CFC could be applied to other hippocampal-dependent associative and reference memory, we carried out other tasks, respectively, including social recognition, novel object recognition, and three-trial CFC. Knockdown of α2-chimaerin caused rapid memory decay of social memory (Fig. 5a) and object recognition memory (Fig. 5b). However, the retention of 24 h fear memory after three-trial CFC remained normal (Supplementary Fig. 2f). This observation, combined with data relevant to α2-chimaerin expression in three-trial CFC (Supplementary Fig. 2e), suggest that α2-chimaerin did not exert effects on memory induced by multiple-trial CFC. Together, these findings identify α2-chimaerin as a memory maintenance molecule in multiple types of hippocampal-dependent memories.

**Regulation of α2-chimaerin modulates the stability of LTP**. To gain further insights into the role of α2-chimaerin in memory enhancement, we recorded long-term potentiation (LTP) induced via a single tetanus (weak stimulation) at the Schaffer collaterals of the hippocampal CA1 region. Knockdown of α2-chimaerin in slices expressing AAV-α2 shRNA led to a reduction of potentiation and accelerated LTP decay (Fig. 6a). In contrast, over-expression of α2-chimaerin in slices expressing AAV-α2-chimaerin induced a marked increase of potentiation and a more stable LTP that lasted over 2 h (Fig. 6b). This observation is consistent with the effects of Rac1 activity manipulation[15], even though induction of LTP is slightly changed by knockdown and overexpression of α2-chimaerin. Thus, LTP stability is highly regulated by α2-chimaerin in a pattern similar to memory stability. This role of α2-chimaerin explains why single-trial CFC-induced memory decays from day 7 to day 14 (Fig. 1a), because the expression of α2-chimaerin falls from day 7 to day 11 (Fig. 3a) while activated Rac1 remains at day 11 (Fig. 1b).

**Discussion**
The current work put forward the concept of reversible forgetting-based dynamic memory maintenance. Specifically, memory is maintained at an intermediate level, which could be either enhanced or suppressed within a defined range (Fig. 7) during the maintenance stage. Such regulation is mediated by the

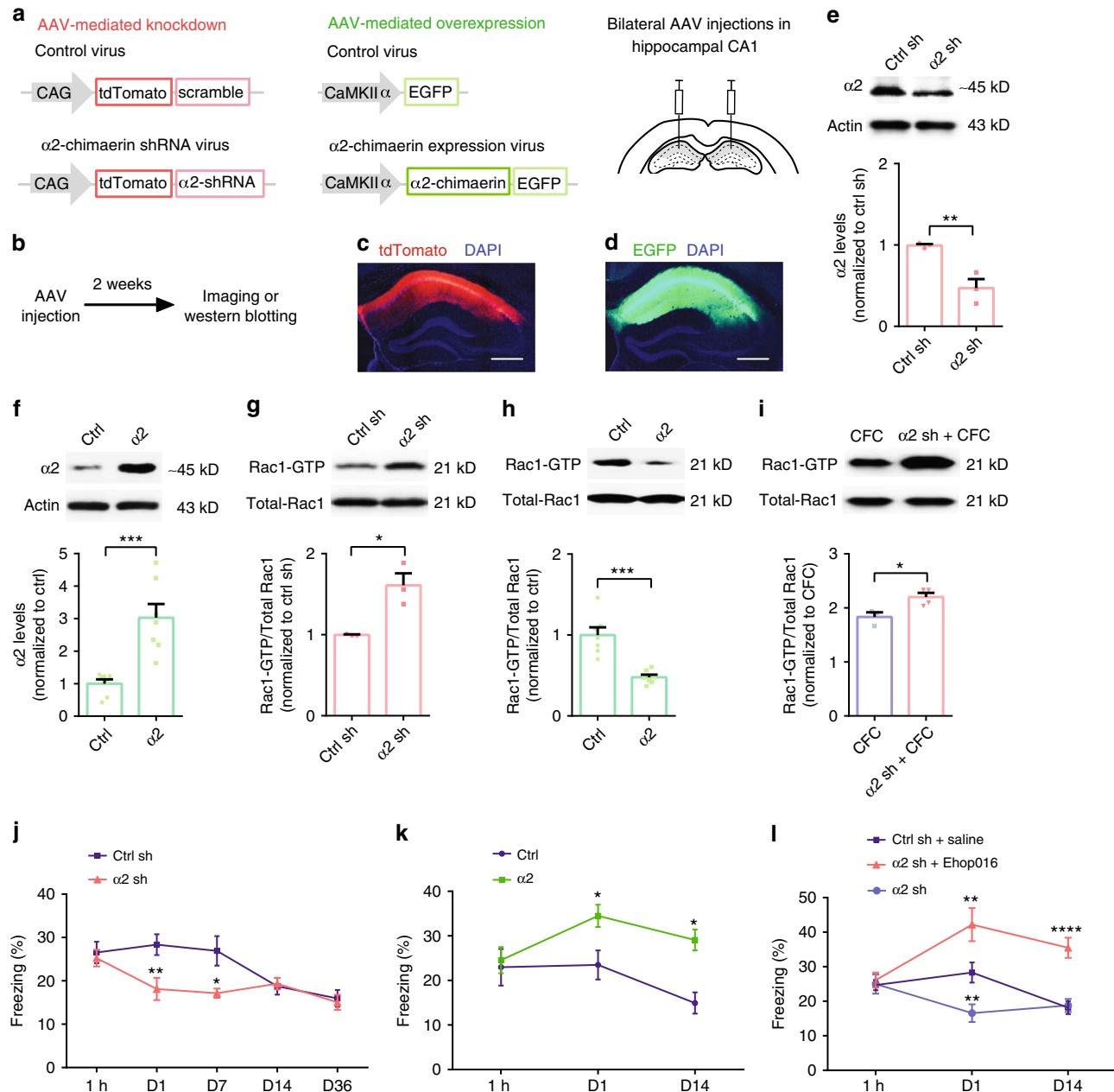

**Fig. 4** Interplay between synthesis of α2-chimaerin and Rac1 activity regulates dynamic memory maintenance. **a** Construction of AAVs and illustration of virus injection sites. **b** Experimental schedule. **c**, **d** Coronal section of the dorsal hippocampus with the viral injections and anti-nuclei (blue). Scale bar, 500 μm. Bilateral AAV injection in dorsal hippocampus results in specific expression of either tdTomato or EGFP in dorsal hippocampus. **e**–**h** Immunoblotting (top) and data (bottom) showing hippocampal α2-chimaerin and Rac1 activity in mice injected with AAV-α2 shRNA (α2 sh) or AAV-α2-chimaerin (α2), as well as their respective controls. *$P < 0.05$, **$P < 0.01$, and ***$P < 0.001$ (from unpaired $t$-test); $n = 3$, 3 mice (**e**, **g**), $n = 7$, 7 mice (**f**, **h**). **i** Immunoblotting (top) and data (bottom) illustrating hippocampal Rac1 activity of CFC mice and trained α2 sh (α2 sh + CFC) mice at 1 day after training. *$P < 0.05$ (from unpaired $t$-test); $n = 3$, 4 mice. **j**–**l** Time courses of freezing in the ctrl sh (AAV-ctrl shRNA), α2 sh, ctrl (AAV-ctrl), α2, ctrl sh with saline treatment (ctrl sh + saline), α2 sh with Ehop016 injection (α2 sh + Ehop016), and α2 sh groups. *$P < 0.05$, **$P < 0.01$, and ****$P < 0.0001$ (from two-way ANOVA); $n = 7$, 8, 15, 13, 11, 11, 16, 10, 9, 11 mice (**j**), $n = 11$, 13, 11, 13, 12, 7 mice (**k**), $n = 3$, 10, 10, 8, 12, 12, 5, 7, 9 mice (**l**). Data are presented as means ± SEM. Also see Supplementary Table 1.

modulation of reversible Rac1-dependent forgetting or expression of a memory molecule α2-chimaerin. We found that single-trial CFC led to sustained Rac1 activation from day 1 to day 11 (Fig. 1b, c and Supplementary 1b), which is capable of causing forgetting, as demonstrated by the effects of AAV-mediated expression of constitutively active Rac1 (Rac1-CA) (Fig. 1d) and optogenetic activation of PaRac1 (Fig. 2g). This forgetting is reversible through inhibition of Rac1 activity via either injection

of a Rac1 inhibitor Ehop016 (Figs. 1e, 2h, j, 4l) or expression of α2-chimaerin (Fig. 4k). The CFC was also capable of stimulating the synthesis of α2-chimaerin that acts as a memory molecule in multiple types of hippocampal-dependent memories through inactivating Rac1 activity (Figs. 3–5). The role is consistent with its effect in the maintenance of LTP (Fig. 6), as opposing to accelerated decay by Rac1 activation[15]. Thus, interplay between activated Rac1 and expressed α2-chimaerin defines the

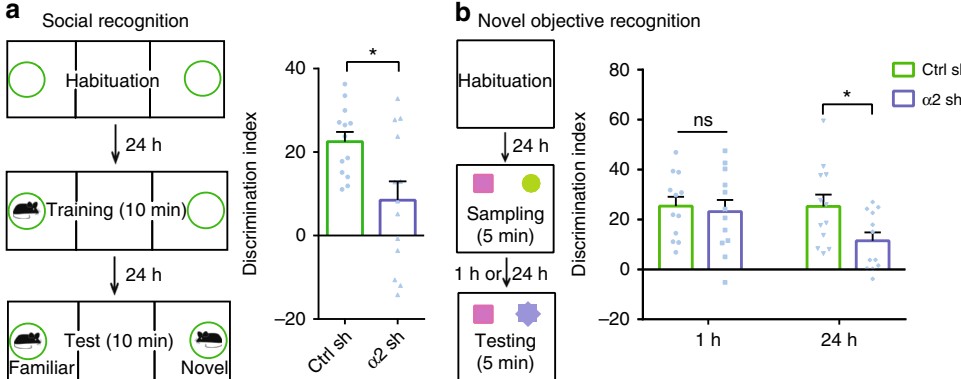

**Fig. 5** Knockdown of α2-chimaerin caused rapid decay of social memory and object recognition memory. **a**, **b** Depiction of the experimental design for social recognition and novel object recognition, respectively (left). **a** Decay of the 24 h social discrimination memory was induced by AAV-α2 shRNA injection (right). *$P < 0.05$ (from unpaired $t$-test); $n = 13$, 13 mice. **b** Effects of α2-chimaerin knockdown on 1 and 24 h object recognition memory (right). *$P < 0.05$ (from two-way ANOVA), $n = 12$, 12 mice. Data are presented as means ± SEM. Also see Supplementary Table 1.

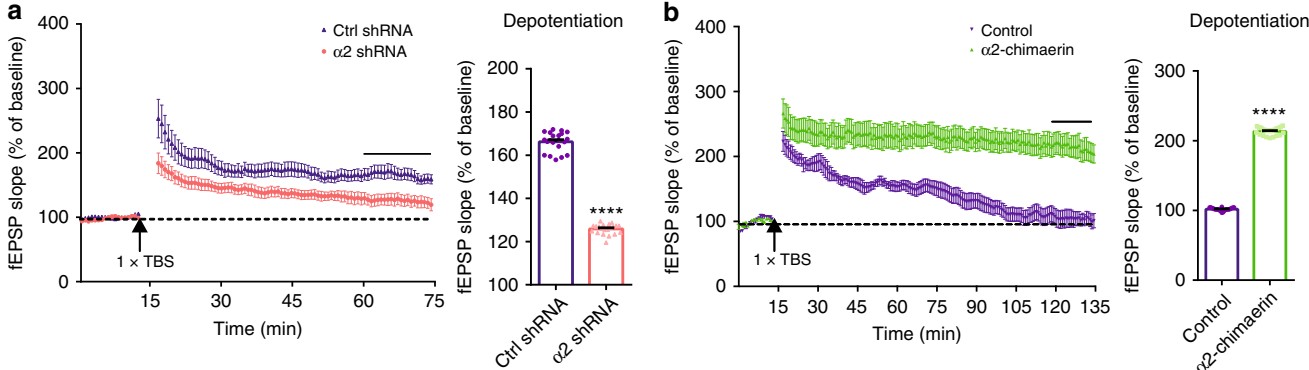

**Fig. 6** α2-chimaerin knockdown and overexpression modulate the stability of LTP. **a**, **b** LTP in slices from α2 sh mice and α2 mice, as well as their controls (ctrl sh or ctrl, respectively) (left), and the average fEPSP slope in the last 15 min of the LTP recording (right). One theta burst stimulation was given 15 min after baseline recording and LTP was recorded in the Schaffer collateral pathway for 1 h (**a**) or 2 h (**b**). ****$P < 0.0001$ (from unpaired $t$-test); $n = 8$, 9 mice (**a**), $n = 11$, 8 mice (**b**). Data are presented as means ± SEM. Also see Supplementary Table 1.

intermediate level of memory during maintenance period and allows dynamic memory maintenance (Fig. 7). Taken together, all of the lines of evidence presented support the novel concept of reversible forgetting-based dynamic memory maintenance.

Recent progress has begun to unravel multiple mechanisms underlying active forgetting, including through activation of Rac1 or Cdc42, as well as through neurogenesis[11]. It however remains largely open for cognitive nature and cellular basis of active forgetting, in which whether active forgetting for memory erasure or for impairment of memory retrieval is not undetermined. A recent study of infantile forgetting in mice suggests that neurogenesis-based forgetting does not erase memory, but decreases memory accessibility[17]. Here we showed that Rac1-dependent forgetting is also resulting from a decrease in memory accessibility through an increase in Rac1 activity, while this inaccessibility can be reversed by inhibition of Rac1 activity via either expression of α2-chimaerin or injection of a Rac1 activity inhibitor (Figs. 1, 2, 4 and 5), suggesting that Rac1-dependent forgetting is reversible in memory trace. This reversible forgetting-based dynamic memory maintenance may not be confined to single-trial CFC-induced contextual fear memory. For the hippocampal-dependent tasks examined, including social recognition (Fig. 5a)[16] and novel object recognition[15] (Fig. 5b), memory decay is all regulated through Rac1-dependent active forgetting.

Thus, any experiences that lead to modulation of Rac1 activity in hippocampus would affect memory accessibility or memory retrieval. Such reversible forgetting-based dynamic memory maintenance might be important for behavior flexibility as well as provide possible mechanisms for symptoms associated with neurological disorders. For instance, study of Alzheimer's disease model suggests that memory loss results from a retrieval problem[33]. It is interesting to investigate whether such a retrieval problem is resulted from enhanced active forgetting.

The enduring efforts in the study of memory maintenance have been in search for molecular mechanisms which permit a LTM to persist[5]. Indeed, there are a number of memory molecules, such as CAMKII, CPEB-3, PKMζ, and brain-derived neurotrophic factor reported in association with memory maintenance[5,6,8,9,34–36]. Here we show that one-trial CFC leads to long-lasting protein synthesis of a memory molecule α2-chimaerin from day 1 to day 7 (Fig. 3), the duration matches well with memory retention curve. In fact, this work also provides a mechanism for its role in maintaining memory, i.e., through inhibition of Rac1-dependent forgetting. Therefore, the memory persists as long as α2-chimaerin expressed.

The translational regulators CPEB proteins are a family of four paralogs: CPEB-1, 2, 3, and 4[8,37]. It is intriguing to note that the CPEB-2 is capable of regulating expression of receptor EphA4[37], which phosphorylates α2-chimaerin in leading to inhibition of

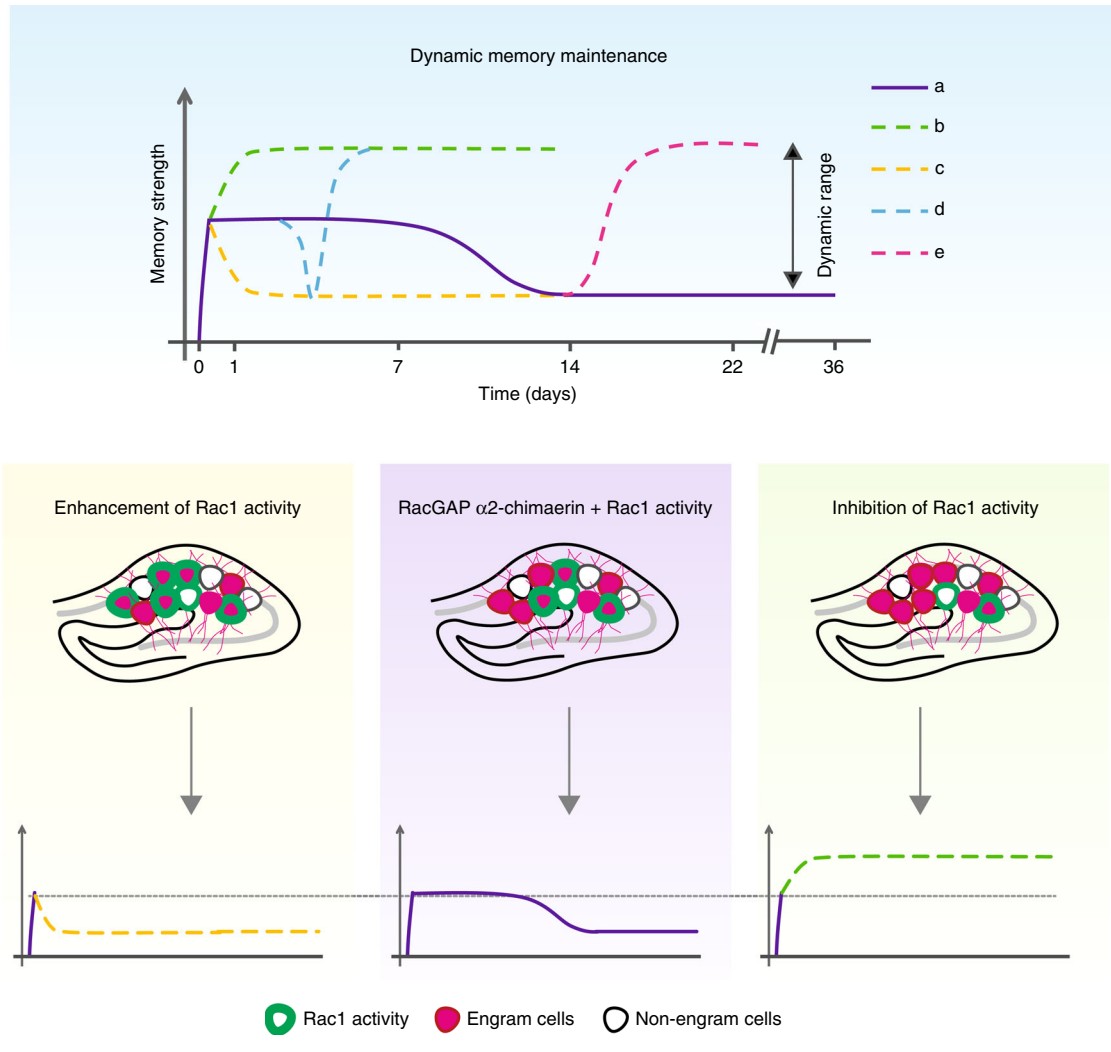

**Fig. 7** Model of reversible forgetting-based dynamic memory maintenance. **a** Intermediate-level memory or normal memory. Memory is maintained at an intermediate level, which is defined by interplay between learning-evoked activated Rac1 and learning-induced expression of α2-chimaerin. **b** The top level of dynamic range. Inhibition of Rac1-dependent forgetting enhances consolidated memory through α2-chimaerin overexpression, AAV-mediated expression of dominant-negative Rac1 (Rac1-DN), or injection of a Rac1 inhibitor Ehop016. **c** The bottom level of dynamic range. Stimulation of Rac1-dependent forgetting suppresses memory through α2-chimaerin knockdown, AAV-mediated expression of constitutively active Rac1 (Rac1-CA) or optogenetic activation of PaRac1. **d**, **e** Both optogenetically induced and naturally forgotten memory could be recovered through injection of a Rac1 inhibitor.

Rac1 activity[18,19,21]. However, there is no report for how expression of α2-chimaerin is regulated and while CPEB-2 or -3 may regulate its expression[37,38]. It would be of interest to determine long-lasting maintenance of Rac1 activation and α2-chimaerin expression induced by learning.

## Methods

**Animals**. C57BL/6J (age: 3–4 months, male) were purchased from the Vital River Laboratory (Animal Technology, Beijing, China) and were housed in groups under standard conditions according to the Tsinghua University animal facility. Mice were maintained on a 12 h light/dark cycle and were tested during the light phase of the cycle. All animal work complied with ethical regulations for animal testing and research, and was done in accordance with Institutional Animal Care and Use Committee (IACUC) approval by Tsinghua University and followed by all Association for Assessment and Accreditation of Laboratory Animal Care (AAALAC) guidelines.

**DNA constructs**. All plasmids were constructed by standard molecular biology procedures and subsequently verified by double-strand DNA sequencing. The α1-chimaerin and α2-chimaerin were synthesized by the GENEWIZ company (sequence are showed below). The pAAV-CaMKIIα-EGFP was a gift from Bryan Roth (Addgene plasmid #50469). The α1-chimaerin and α2-chimaerin were sub-

cloned into AAV backbone pAAV-CaMKIIα-EGFP using the BsrGI/EcoRI restriction enzymes.

α1-chimaerin (Sequence ID: NM_001206602.1)

5′-ATGCCATCCAAAGAGTCTTGGTCAGGGAGGAAAACTAATAGGGCT
GCAGTTCACAAATCAAAACAAGAGGGCCGTCAGCAAGATTTATTGATAG
CAGCCTTGGGAATGAAACTGGGTTCTCCAAAGTCGTCTGTGACAATCTG
GCAACCTCTGAAACTCTTTGCTTATTCGCAGTTGACATCACTTGTTAGAA
GAGCAACTCTGAAAGAAAACGAGCAAATTCCAAAATATGAAAAGATTCA
CAATTTCAAGGTGCATACATTCAGAGGGCCACACTGGTGTGAATACTGT
GCCAACTTTATGTGGGGTCTCATTGCTCAGGGAGTGAAATGTGCAGATT
GTGGTTTGAATGTTCATAAGCAGTGTTCCAAGATGGTCCCAAATGACTG
TAAGCCAGACTTGAAGCATGTCAAAAAGGTCTACAGCTGTGACCTTACG
ACGCTCGTGAAAGCACATACCACTAAGCGGCCAATGGTGGTAGACATGT
GCATCAGGGAGATTGAGTCTAGAGGTCTTAATTCTGAAGGACTATACCG
AGTATCAGGATTTAGTGACCTAATTGAAGATGTCAAGATGGCTTTCGAC
AGAGATGGTGAGAAGGCAGATATTTCTGTGAACATGTATGAAGATATCA
ACATTATCACTGGTGCACTTAAACTGTACTTCAGGGATTTGCCCAATTCCA
CTCATTACATATGATGCCTACCCTAAGTTTATAGAATCTGCCAAAATTAT
GGATCCGGATGAGCAATTGGAAACCCTTCATGAAGCACTGAAACTACTG
CCACCTGCTCACTGCGAAACCCTCCGGTACCTCATGGCACATCTAAAGA
GAGTGACCCTCCACGAAAAGGAGAATCTTATGAATGCAGAGAACCTTGG
AATCGTCTTTGGACCCACCCTTATGAGATCTCCAGAACTAGACGCCATG
GCTGCATTGAATGATATACCGTATCAGAGACTGGTGGTGGAGCTGCTTA
TCAAAAACGAAGACATTTTTATTTTAA-3′

α2-chimaerin (Sequence ID: NM_001822.5)

5′-ATGGCCCTGACCCTGTTTGATACAGATGAATATAGACCTCCTGTTT
GGAAATCTTACTTATATCAGCTACAACAGGAAGCCCCTCATCCTCGAAG
AATTACCTGTACTTGCGAGGTGGAAAACAGACCAAAGTATTATGGAAGA
GAGTTTCATGGCATGATCTCCAGAGAAGCAGCCGACCAGCTCTTGATTG
TGGCTGAGGGGAGCTACCTCATCCGGGAGAGCCAGCGGCAGCCAGGGA
CCTACACTTTGGCTTTAAGATTTGGAAGTCAAACCAGAAACTTCAGGCT
CTACTACGATGGCAAGCACTTTGTTGGGGAGAAACGCTTTGAGTCCATC
CACGATCTGGTGACTGATGGCTTGATTACTCTCTATATTGAAACCAAGGC
AGCAGAATACATTGCCAAGATGACGATAAACCCAATTTATGAGCACGTA
GGATACACAACCTTAAACAGAGAGCCAGCATACAAAAAACATATGCCAG
TCCTGAAAGAGACAACATGATGAGAGAGATTCTACAGGCCAGGATGGGG
TGTCAGAGAAAAGGTTGACATCACTTGTTAGAAGAGCAACTCTGAAAGA
AAAACGAGCAAATTCCAAAATATGAAAAGATTCACAATTTCAAGGTGCAT
ACATTCAGAGGGCCACACTGGTGTGAATACTGTGCCAACTTTATGTGGG
GTCTCATTGCTCAGGGAGTGAAATGTGCAGATTGTGGTTTGAATGTTCA
TAAGCAGTGTTCCAAGATGGTCCCAAATGACTGTAAGCCAGACTTGAAG
CATGTCAAAAAGGTCTACAGCTGTGACCTTACGACGCTCGGTGAAAGCAC
ATACCACTAAGCGGCCAATGGTGGTAGACATGTGCATCAGGGAGATTGA
GTCTAGAGGTCTTAATTCTGAAGGACTATACCGAGTATCAGGATTTAGT
GACCTAATTGAAGATGTCAAGATGGCTTTCGACAGAGATGGTGAGAAGG
CAGATATTTCTGTGAACATGTATGAAGATATCAACATTATCACTGGTGCA
CTTAAACTGTACTTCAGGGATTTGCCAATTCCACTCATTACATATGATGC
CTACCCTAAGTTTATAGAATCTGCCAAAATTATGGATCCGGATGAGCAA
TTGGAAACCCTTCATGAAGCACTGAAACTACTGCCACCTGCTCACTGCG
AAACCCTCCGGTACCTCATGGCACATCTAAAGAGAGTGACCCTCCACGA
AAAGGAGAATCTTATGAATGCAGAGAACCTTGGAATCGTCTTTGGACCC
ACCCTTATGAGATCTCCAGAACTAGACGCCATGGCTGCATTGAATGATA
TACGGTATCAGAGACTGGTGGTGGAGCTGCTTATCAAAAACGAAGACAT
TTTATTTTAA-3′

The U6 promoter-driven α1-chimaerin shRNA (α1 shRNA), α2-chimaerin shRNA (α2 shRNA), or negative control scrambled shRNA (ctrl shRNA) constructs were synthesized (GENWIZ company) and sub-cloned into AAV backbone pAVV-CAG-tdTomato (Addgene plasmid # 59462) using the EcoRI/HindIII and EcoRV/HindIII restriction enzymes, respectively. The α1-chimaerin shRNA sequence binding to the mouse *α1-chimaerin* was 5′-GCTT TCAGCAATGTGTCAT-3′[39]. The α2-chimaerin shRNA sequence binding to the mouse *α2-chimaerin* was 5′-GCACATGGCAGTCCTGAAA-3′, whereas the negative control scrambled shRNA (ctrl shRNA) was 5′-GTTCATATGTTCACCTATT-3′[22]. The U6 promoter sequence was 5′- GATC CGACGCCGCCATCTCTAGGCCCGCGCCGGCCCCCTCGCACAGACTTGTG GGAGAAGCTCGGCTACTCCCCTGCCCCGGTTAATTTGCATATAATATTT CCTAGTAACTATAGAGGCTTAATGTGCGATAAAAGACAGATAATCTGTT CTTTTTAATACTAGCTACATTTTACATGATAGGCTTGGATTTCTATAAGA GATACAAATACTAAATTATTATTTTAAAAAACAGCACAAAAGGAAACTC ACCCTAACTGTAAAGTAATTGTGTGTTTTGAGACTATAAATATCCCTTGG AGAAAAGCCTTGTTT-3′.

**Antibodies and reagents**. The following antibodies were used: primary antibodies for α1-chimaerin and α2-chimaerin were raised in rabbits immunized with keyhole limpet hemocyanin-coupled synthetic peptides as previously reported[39]. Keyhole limpet hemocyanin-coupled synthetic peptides for α1-chimaerin was MPSKESWSGRKANR and keyhole limpet hemocyanin-coupled synthetic peptides for α2-chimaerin was HDEKEATGQDGVSEKR. Anti-Rac1-GTP (active Rac1) monoclonal antibody was from the NewEast company (Cat. No. 26903). Anti-Rac1 antibody was from BD Transduction Laboratories (Cat. No. 610650). Anti-phospho-PAK1/2/3 antibody was from novusbio company (Cat. No. NB100-82131). Anti-Cleaved Caspase-3 antibody was from Cell Signaling (Cat. No. 9661). Horseradish peroxidase (HRP)-conjugated goat anti-mouse IgG (Cat. No. #7072) and HRP-conjugated goat anti-rabbit IgG (Cat. No. #7071) were from Cell Signaling Technology. Alexa Fluro® 488 affinipure donkey anti-mouse IgM was from the Jackson ImmunoResearch Laboratories company (Cat. No. #711-545-140). Actin antibody was from the Merck Millipore company (Cat. No. #MAB1501).

**Drug**. Ehop016 (Shanghai Sun-shine Chemical Technology Co., Ltd) was dissolved in a solution containing 1% dimethylsulfoxide/30% PEG/1% Tween-80. Mice were intraperitoneally injected with 20 mg per kg of Ehop016 solution or an equivalent volume of saline. At the dose used, Ehop016 has high efficiency to block Rac1 activity[15]. ANI (Selleck, Cat. No. S7409) was dissolved in 0.9% saline and the pH was adjusted with 1 N HCl to 7.0–7.4. Mice were subcutaneously injected with 150 mg per kg of ANI or an equivalent volume of saline immediately after training. This amount of ANI was shown to effectively inhibit cerebral protein synthesis in mice (~96%)[40].

**Surgery and viral injections**. Mice were anesthetized with 0.2% sodium pentobarbital (5 ml/kg) and fixed to a stereotaxic frame. Their body temperature was kept at 36 ℃ by a heating pad and their skull was exposed. Furthermore, holes in the skull were drilled by a micromotor drill. The injection of the virus was performed using a 10 ml nanofil syringe under controlled by the UMP3 and Micro4 system (WPI), with a speed of 50 nl/min. The dorsal hippocampus

injections were bilaterally targeted to −2.0 mm AP, ±1.5 mm ML, and −1.5 mm DV; the DG injections were bilaterally targeted to -2.0 mm AP, ±1.5 mm ML, and −2.0 mm DV. The viral volumes of the AAV-Rac1-CA, AAV-Rac1-DN, and AAV-control were 400 nl for DG and 600 nl for CA1. Mice were bilaterally injected with 1 µl of either AAV-α1-chimaerin, AAV-α2-chimaerin, AAV-control, AAV-α1-chimaerin shRNA, AAV-α2-chimaerin shRNA, or AAV-control shRNA. For labeling of the CA1 engram cells, a virus cocktail AAV9-c-fos:tTA (300 nl of AAV9-c-fos:tTA) and 300 nl of the AAV9-TRE-tdTomato into the CA1 (-2.2 mm AP, ±1.7 mm ML, and −1.6 mm DV). For the activation of Rac1 activity in CA1 engram cells, a virus cocktail AAV9-c-fos:tTA (300 nl of AAV9-c-fos:tTA and 300 nl of AAV9-TRE:photoactivatable Rac1 (PaRac1)-tdTomato[30,31,41] or the AAV9-c-fos-EGFP (as the control group) were injected into the mice CA1 (-2.2 mm AP, ±1.8 mm ML, and −1.6 mm DV). Following the injections, the needle stayed for 10 min before slowly withdrawn and the wound sutured. After surgery, animals were allowed to recover for two weeks prior to the performance of all subsequent experiments.

**Optical fiber implant**. Mice were bilaterally injected with a virus cocktail (500 nl of AAV9-c-fos:tTA) and 500 nl of the AAV9-TRE:PaRac1-tdTomato[30,31,41] or the AAV9-c-fos-EGFP (as the control group) into the CA1. Furthermore, they were bilaterally implanted with optical fibers (200 µm core, 0.37 numerical aperture) into the CA1 (−2.2 mm AP, ±1.8 mm ML, and −1.5 mm DV). Mice were allowed to recover for two weeks under the ON-Dox condition prior to the start of the behavioral experiments. The cohorts of mice used for the optogenetic engram stimulation, mice were fed regular food, without Dox, for 24–30 h until contextual fear-conditioning training. After CFC, mice were returned to their home cage under the ON-Dox condition. Following behavioral testing, brain sections were examined to confirm efficient virus-mediated labeling of the target areas.

**In vivo photoactivation of Rac1**. Mice were intra-CA1 implanted with optic fiber after AAV injection. The optic fibers of mice were connected to a 473 nm blue laser diode via a FC/PC adaptor. Light intensity (~15 mW/mm$^2$) was measured at the tip of the fiber. A blue light from a laser was ON for 30 s to stimulate the photo-activatable Rac1 for five trials, with trial intervals of 120 s[31]. For all behavioral procedures, the freezing behavior was measured in light stimulated mice at 1.5 h after stimulation. Successively, animals were put back to their home cage.

**Contextual fear conditioning**. The single- and three-trial CFC procedures were based on those of Denny et al.[42]. The fear-conditioning test was conducted using the HABITEST Modular Behavioral Test System. A Coulbourn Habitest chamber (27 cm × 28 cm × 30.5 cm) had a stainless-steel rod floor which was connected to a shock generator in a sound-attenuating box. In single-trial CFC sessions, mice were individually placed in the conditioning area and freely explored the area for 3 min. Then, mice were exposed to one footshock (2 s, 0.8 mA) and were returned to their home cage 30 s after. In the three-trial CFC task, the footshock (2 s, 0.8 mA) was delivered at 180, 240, and 300 s. Mice remained in the conditioning chamber for a total of 330 s. In only context or only shock training, mice were exposed in a context for 3 min without shock or one footshock (2 s, 0.8 mA) in the absence of context. After that, mice were placed back in their home cage. During testing, mice were placed back in the conditioning chamber for 4 min at different times (1 h, days 7, 14, or 36). For all procedures, the animal freezing behaviors were monitored using a manufacturer's software.

**Social discrimination test in the three-chamber apparatus**. Subject mice were individually placed in the three-chamber apparatus for 10 min to habituate to the environment (day 1). On day 2, a 6-week-old male C57BL/6J mouse was placed in the left grid enclosure for familiarization. Successively, the test mouse was located in the center compartment of the three-chamber apparatus for 10-min social training. During experimental testing (1 h and day 3), a novel 6-week-old male C57BL/6J mouse was placed in the right grid enclosure and the familiar mouse in the left grid enclosure. And then, test mouse was placed into the social chamber and was allowed to explore for 10 min. The amount of time for the test mouse spent in close interaction was recorded using the Any maze system. The discrimination index (DI) was calculated using the following formula: (time exploring the novel mouse − time exploring the familiar mouse)/(time exploring the novel mouse + time exploring the familiar mouse) * 100.

**Novel object recognition task**. For the habituation phase, individual adult male mice were placed in a chamber (50 cm × 50 cm × 40 cm) and allowed to freely explore the context for 10 min while being recorded by an overhead camera. This phase was also conducted as the open field test to assess the animal's basal locomotor activity indicated by the average speed and anxiety levels indicated by the percentage of time spent in the middle arena (30 cm × 30 cm). For the sampling phase, the animal was placed in the same chamber containing two different objects for 5 min and allowed to explore the object. To saturate the odor left by the previous mice and to make the mice more relaxed, we placed standard animal beddings in the chamber[15]. For the testing phase, one of the objects was exchanged with a new one, and the time spent exploring the two objects was separately recorded using ANY-MAZE software. The DI was calculated using the following

formula: (time exploring the novel object − time exploring the familiar object)/ (time exploring the novel object + time exploring the familiar object) × 100. For the memory decay curve, time intervals between sampling and testing were 1 and 24 h.

**Western blotting**. Isolated hippocampi were homogenized in cell lysis buffer (Beyotime, Cat. No. P0013) with protease inhibitors. The protein concentration was determined using the BCA protein assay kit (Beyotime, Cat. No. P0012). For the detection of the levels of α1-chimaerin, α2-chimaerin, total Rac1 and actin, 20 μg of homogenates was separated by 15% SDS-polyacrylamide gel electrophoresis and transferred to nitrocellulose membranes (Pall Corporation, Cat. No. 66485). Membranes were blocked in milk solution (5% milk in TBS and 0.1% Tween 20) for one hour at room temperature. Subsequently, membranes were individually incubated with primary antibodies against α1-chimaerin (anti-α1-chimaerin polyclonal antibody, 1:2000, rabbit), α2-chimaerin (anti-α2-chimaerin polyclonal antibody, 1:1000, rabbit), total Rac1 (1:2000, rabbit), and actin (1:1000, mouse) overnight at 4 °C. All the HRP-conjugated secondary antibodies were used at 1:2000 dilutions for membranes incubation at room temperature for one hour. For the relative level of the Rac-GTP assay, a GST-tagged PAK-PBD beads (Cytoskeleton, Cat. No. #PAK02) were incubated with the lysate mix (300 μg) for overnight at 4 °C. The beads were washed three times with lysis buffer at room temperature. Then, bound proteins were eluted with loading buffer (Beyotime, Cat. No. P0015B) and analyzed by western blotting. Image quantification analysis in bands of the western blots was calculated by the ImageJ software (National Institutes of Health).

**Immunofluorescence**. Mice were anesthetized with 0.2% sodium pentobarbital (5 ml/kg) and perfused intracardially with 30 ml of 4% paraformaldehyde. The brains were fixed in 4% paraformaldehyde overnight at 4 °C and serial coronal sections (30 μm) were then taken throughout the hippocampus using a vibratome. The floating sections were collected and rinsed in a 0.1 M phosphate-buffered saline (PBS) (pH 7.4) solution with 0.1% Triton X-100 and incubated with blocking solution (10% donkey serum and 0.1% Triton X-100) for 2 h at room temperature. The slices were then incubated with the following primary antibodies at 4 °C for overnight: anti-active Rac1 (1:500, mouse), anti-Cleaved Caspase-3 (1:400, rabbit), and anti-phospho-PAK1/2/3 (1:200, rabbit). Following three times rinsed (30 min each) in the PBS solution, slices were incubated with the secondary antibodies conjugated with dyes. Then sections were washed three times with PBS solution and mounted on slides with antifade mounting medium (VECTASHIELD, Cat. No. H-1200). The images of the immunohistochemistry were captured on a Zeiss LSM 710.

**Cell counting**. Cell counting of c-Fos-positive cells and Rac1-active cells was performed in the CA1 by utilizing imaging analysis function of Zeiss software (Zen blue 2.3). The quantification of the number of c-Fos-positive cells was performed by thresholding c-Fos immunoreactivity above background levels automatically. The quantification of the number of Rac1-GTP cells was performed by setting the threshold manually from 1000 to 4096. The quantification of the number of phospho-PAK cells was performed by setting the threshold manually from 692 to 4096. The quantification of the number of cells with 4′,6-diamidino-2-phenylindole was performed by setting the threshold manually from 616 to 4096.

**Preparation of the slices and electrophysiological recordings**. Transverse hippocampal slices were prepared from 4-month-old mice. Animals were killed by decapitation in accordance to the institutional regulations. Hippocampi were sliced (300 μm) by a VF-300 microtome (Precisionary Instruments) and kept in an interface chamber with oxygenated artificial cerebrospinal fluid (ACSF; 124 mM NaCl, 3 mM KCl, 1.25 mM $KH_2PO_4$, 1 mM $MgSO_4$, 2 mM $CaCl_2$, 26 mM $NaHCO_3$, and 10 mM glucose) for at least 1 h. Following the recovering at room temperature, the slices were transferred to the field excitatory post-synaptic potentials (EPSPs) recording chamber (MED-PG515A, Alpha MED Science) with ACSF (flow rate 1.5 ml/min) for at least 20 min prior to the recording. Filed EPSPs were evoked from CA3-CA1 synapses through the stimulation of the Schaffer collateral axons and recorded in CA1 extracellular fEPSPs. The stimulation intensity was adjusted to elicit a 30% maximal response. The LTP protocol for the 100 Hz consisted in 10 × 100 Hz bursts (four pulses per burst) with a 200 ms interval between bursts. Data acquisition and analysis were performed using the multielectrode MED64 hardware and software packages (Panasonic).

**Statistics**. Statistical analyses were performed in GraphPad Prism. All data were analyzed with an unpaired $t$-test, one-way analysis of variance (ANOVA) or two-way ANOVA where appropriate. The data are shown as the mean ± SEM and NS indicates non-significance ($p > 0.05$). The significant levels were set to $P = 0.05$. Significant for comparison: *$p < 0.05$; **$p < 0.01$; ***$p < 0.001$; ****$p < 0.0001$. The specific statistical tests used and the details of $p$-values for each experiment can be found in Supplementary Table 1.

**Reporting summary**. Further information on research design is available in the Nature Research Reporting Summary linked to this article.

## Data availability

All pertinent data generated or analyzed during this study are included in this published article and its accompanying Supplementary Information files.

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

## Acknowledgements

We greatly thank Professor Peter Scheiffele at University of Basel for providing an anti α2-chimaerin antibody. We thank L.Z. for help with the experiments and all the members of the Zhong lab for their support. This work was supported by grants from the National Science Foundation of China (91632301 to Y.Z.), the Beijing Municipal Science and Technology Commission (Z161100002616010 to Y.Z.) and (Z161100000216132/01 to Y.Z.), the Peking University-Tsinghua University-National Institute Biological Science Joint Graduate Program and the Tsinghua-Peking Joint Center for Life Sciences.

## Author contributions

L.L. and Y.Z. designed the study. L.L., J.X., and Y.L. conducted the viral behavioral experiments. L.L., J.X., and Y.L. performed all electrophysiological experiments. L.L. and Y.W. performed and analyzed all imaging data. L.L., J.Z., and Y.W. performed the optogenetic behavioral experiments. The western blot data were performed and analyzed by L.L. and Q.L. The molecular cloning was conducted by Y.L. The manuscript was written by L.L. and Y.Z.

## Competing interests

The authors declare no competing interests.

## Additional information

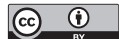

