## [Transparent Peer Review File · Nature Communications]

Reviewers' comments:

Reviewer #1 (Remarks to the Author):

The study investigates the roles of Rac1 and RacGAP α 2-chimaerin in memory maintenance and how affecting their activities can lead to forgetting or on the other hand to memory enhancement. The authors reveal that inhibition of learning-induced Rac1-dependent forgetting in hippocampus leads to enhancement of contextual fear memory. They have further show that memory maintenance is dynamic and can be regulated by reversible Rac1-dependent activity within CA1 memory engram cells. In addition, they show that α 2-chimaerin RacGAP activity and level are involved in the maintenance of contextual fear conditioning where- its inhibition suppresses memory and its activation enhances memory. α 2-chimaerin knockdown and overexpression modulate the stability of long-term potentiation.

The results here are novel but do not introduce new concepts into the field of memory maintenance as the role of Rac1 in modulating forgetting and synaptic plasticity have been shown already by this group in other elegant studies. However, they extend in this study the results to an additional behavioral paradigm (contextual fear conditioning) and introduced the involvement of α 2-chimaerin in this process.

I have additional comments:

- 1) How do they know that multiple injection of Rac1 inhibitor do not cause permanent damage that may lead to an enhancement? For example progressive cell death may lead to changes in memory? How they determine the protocol? It should be determined that no damage in tissue is observed after multiple injections.
- 2) Same comment as above- How do they know that constitutive activation or inhibition of Rac1 with Rac1-DN or Rac1-CA do not cause permanent non-specific damage that affects the ability to form memory (enhance or inhibit)? Is it specific to memory?
- 3) Regarding figure 2C - What is strong immunohistochemistry what is medium? Some controls are missing to determine that these are learning-specific cells such as shock or immediate context only.
- 4) Figures 2C-D the distribution of TRE-paRac1-dtTomato after training looks very different from the distribution of with tomato only? Why is that? And therefore can tomato only serve as appropriate control if distributed differently?
- 5) Figure 2E- It is not clear- if the memory is back what happened to it during activation of PA-Rac? Did PA-Rac1 erased the engram on the day it was activated and then it returned back spontaneously to the level before activation of PA-Rac1?
- 6) "As an ideal control, manipulating the expression of α 1-chimaerin induced no significant changes in memory strength during the maintenance stage (Figures S7A-S7E) or the relevant behavioral parameters" The sentence is not clear.
- 7) What keeps Rac1 active the whole time to maintain memory? As well as RacGAP.

Reviewer #2 (Remarks to the Author):

In this manuscript, authors examined the roles of hippocampal Rac1-GTP and α 2-chimaerin on memory maintenance after contextual fear conditioning. First, they showed that Rac1-GTP activity

is associated with the maintenance of contextual fear memory, this is consistent with their previous data using object recognition memory (Liu et al., 2016), by using global and engram-specific Rac-GTP manipulation. Authors found that $\alpha 2$ -chimaerin acts on as a crucial suppressor for the Rac1-activity for contextual fear memory by using the viral-mediated both knockdown and overexpression approaches. Consistent with behavioral data, authors also showed that induction of long-term potentiation is modulated by the artificial manipulation of $\alpha 2$ -chimaerin expression. Although this study presents novel molecular mechanisms about how contextual memory is maintained from the aspect of dynamics for the balance between Rac1-GTP and $\alpha 2$ -chimaerin expression, I have some major and minor concerns as described below.

Major points;

- 1) Paper format: title and introduction are vague. Title has to be concrete, this is not review article. In introduction, Authors need to start write with previous papers about roles of Rac1-GTP and $\alpha 2$ -chimaerin on learning and memory and synaptic plasticity.
- 2) Conceptual advance: The manuscript is an extension study from previous their paper (Liu et al., 2016). Fig 1 and 2 showed re-examination with contextual fear conditioning. Fig3 and Fig4 is novel, but analysis is still weak. Author need to more focus on the role of $\alpha 2$ -chimaerin on memory maintenance as whole manuscript.
- 3) Cell counting issue: Overall, cell counting analysis is superficially examined. In Fig1C, is seems all of CA1 cells shows Rac1-GTP immunoreactivity. If so, very strange, because if Rac1-GTP is essential for several type of memory, all memory will be effected. Authors need to quantify. In Fig2C and D, authors need to quantitatively count the cells which are labeled with RAC1-GTP positive in tdTomato+ and tdTomato- cells. In Fig 3, Authors need to examine $\alpha 2$ -chimaerin positive in Ca1 engram+ and engram- cells. They should significantly overlap within same population,
- 4) In Fig 5, Authors tried to show the effect of LTP maintenance for knockdown and overexpression of $\alpha 2$ -chimaerin. However, the manipulation already caused the effect on the induction of LTP. This is inconsistent with their fear conditioning data, since it is considered protein-synthesis dependent LTP is more than 4 hours usually.
- 5) Authors need to show the expression level of $\alpha 2$ -chimaerin when they conduct strong CFC protocol.
- 6) To generalize their finding, authors need to examine knockdown of $\alpha 2$ -chimaerin in strong CFC protocol and also object recognition memory.
- 7) While authors examined the effect of knockdown and over expression of $\alpha 2$ -chimaerin on Rac1-GTP, Authors did not show the effect of inhibition/activation of Rac1-GTP on $\alpha 2$ -chimaerin. Both $\alpha 2$ -chimaerin and Rac1-GTP may mutually regulate their activity or expression level.

Minor points,

- 1) In Figure 4A to D, the authors described as AAV injection into dorsal hippocampus, but specific fluorescent signal of either tdTomato or EGFP seem to be only expressed in CA1 region without other hippocampal subregions. It should be precisely described in the article as knockdown and overexpression of $\alpha 2$ -chimaerin in "hippocampal CA1 region".
- 2) Through this paper, the author frequently uses the term "middle level" memory or that memory is maintained at "middle level" to refer to the state of memory. It should be clearly described what the "middle level" means or should be rephrased with suitable word for this.
- 3) In line 218 of main text, (Figure S6A-S6C) should be "Figure S6A, S6B". The Figure S6C is not existed.
- 4) In line 228 of main text, (Figure S4D) should be "Figure S4E".
- 5) Through discussion at the first paragraph, the authors mention to the main figures within the

sentences to support their experimental evidences, but is so obscurely. It should make sure that which figure supports on your conclusive evidences.

6) In Figure1 legend at (B), native (N) should be (“Na”).

7) In Figure 6, it would be better to put on “(days)” in the x-axis index.

8) Title for Figure S7 “after both the one and three-shock CFC” doesn’t make sense. Which graph is shown in which conditioning procedures?

Reviewer #3 (Remarks to the Author):

In this manuscript by Lv and colleagues, the authors investigate the role of the small G-protein Rac1 in the regulation of long-term memory maintenance. The authors demonstrate using a single trial fear conditioning paradigm that memory stability is regulated by Rac1 that is upregulated following fear conditioning, with a delayed onset of expression. Using a series of pharmacological and genetic strategies, the authors demonstrate that persistent activation of Rac1 impairs long-term memory storage by facilitating forgetting. In general the results of the study are interesting and will be of broad interest, but there are a few experiments that are required to fully support the conclusions. In addition, the manuscript is written with a lot of jargon that makes it difficult to read and will be even more difficult to read by a non-expert. The insistence on the use of forgetting, as opposed to memory maintenance is particularly confusing for the reader. Forgetting is an abstract construct that is difficult to ascribe to the simple animal behavioral paradigm used here and should be avoided.

Specific comments:

1. The authors use a repeated measures assay to assess memory stability. The behavioral assay confounds their analysis because the effects could be mediated by deficits in memory recall and reconsolidation. To validate that their observations are mediated by instability of the initial memory and not deficits in recall/reconsolidation in their initial tests and subsequent experiments the authors need to demonstrate using distinct cohorts of mice for each time point. One cohort for 1 hr, 1 for 1 day, 1 for 7 days, 1 for 14 days, and 1 for 36 days. If their hypothesis is correct, then the memory should be weaker in the 14 day and 36 day cohorts. This would also more directly parallel their histological and western blot analysis.
2. The authors need to demonstrate that either Rac1 is not induced in the 3-trial fear conditioned mice, or that there is a persistent expression of alpha-chimaerin. This would support their claims that “forgetting” is mediated by Rac1 activation which should be not present in mice that have persistent memory.
3. The authors should demonstrate that Rac1 has the opposite effects of alpha-chimaerin on plasticity to fully support their hypothesis.

Reviewer #1

The study investigates the roles of Rac1 and RacGAP α 2-chimaerin in memory maintenance and how affecting their activities can lead to forgetting or on the other hand to memory enhancement. The authors reveal that inhibition of learning-induced Rac1-dependent forgetting in hippocampus leads to enhancement of contextual fear memory. They have further show that memory maintenance is dynamic and can be regulated by reversible Rac1-dependent activity within CA1 memory engram cells. In addition, they show that α 2-chimaerin RacGAP activity and level are involved in the maintenance of contextual fear conditioning where- its inhibition suppresses memory and its activation enhances memory. α 2-chimaerin knockdown and overexpression modulate the stability of long-term potentiation.

The results here are novel but do not introduce new concepts into the field of memory maintenance as the role of Rac1 in modulating forgetting and synaptic plasticity have been shown already by this group in other elegant studies. However, they extend in this study the results to an additional behavioral paradigm (contextual fear conditioning) and introduced the involvement of α 2-chimaerin in this process.

I have additional comments:

- 1) How do they know that multiple injection of Rac1 inhibitor do not cause permanent damage that may lead to an enhancement? For example progressive cell death may lead to changes in memory? How they determine the protocol? It should be determined that no damage in tissue is observed after multiple injections.
- 2) Same comment as above- How do they know that constitutive activation or inhibition of Rac1 with Rac1-DN or Rac1-CA do not cause permanent non-specific damage that affects the ability to form memory (enhance or inhibit)? Is it specific to memory?

Response 1.1-1.2: These two concerns focus on whether permanent non-specific tissue damage is induced for manipulation of Rac1 activity through either multiple injection of Rac1 inhibitor or through expression of Rac1-DN and Rac1-CA. We address this concern from three aspects. Firstly, we performed immunostaining for cleaved caspase-3 (cell death marker¹) in the hippocampal slice. We found multiple injection of Rac1 inhibitor (Ehop016) or the viral manipulations of Rac1 activity (with Rac1-DN or Rac1-CA) did not induce cell death (Supplementary Fig. 3d and Supplementary Fig. 4c, revised manuscript). Secondly, our previous and current studies show that Rac1 manipulations through expression of either AAV-Rac1-DN or AAV-Rac1-CA had no effect on learning and memory formation^{2,3} (Fig. 1a), suggesting that the function of hippocampus with such treatments remains largely normal for learning and memory. Thirdly, accelerated forgetting induced by expression of AAV-Rac1-CA can be mimicked by acute light-stimulation of PaRac1 (Fig. 2g, h, revised manuscript), suggesting observed forgetting is physiologic instead of through tissue damage.

- 3) Regarding figure 2C - What is strong immunohistochemistry what is medium? Some controls are missing to determine that these are learning-specific cells such as shock or immediate context only.

Response 1.3: We classified activated Rac1 (Rac1-GTP) positive cells into total and strong of two groups. The quantification of the total number of Rac1-GTP cells was performed by setting the threshold manually from 1000 to 4096 and the number of strong Rac1-GTP cells was performed by setting the threshold manually from 2307 to 4096. Since there is no further

functional analysis of these two groups and for clarity, we only presented the total number and removed the strong group in revision (Fig. 2e). To address the reviewer's concern of missing control, we now included shock only and context only in revised Fig. 2c and Supplementary Fig. 5a. The additional data support the vast majority of c-fos⁺ cells are learning-specific in CFC group (Fig. 2c, d and Supplementary Fig. 5a, b, revised manuscript).

4) Figures 2C-D the distribution of TRE-paRac1-dtTomato after training looks very different from the distribution of with tomato only? Why is that? And therefore can tomato only serve as appropriate control if distributed differently?

Response 1.4: The visual difference presented in the old version is missing leading. To visualize more precisely the expression pattern, we lowered the upper threshold of histogram in Fig. 2f so that Fig. 2d (old) is much intensive due to overexposed. In fact, there is no significant difference between the distribution of TRE-tdTomato and TRE-PaRac1-tdTomato when the upper threshold is move back to the level similar to that in Fig. 2c. We have now replaced the over exposed image with an appropriate image (Fig. 2f, revised manuscript).

5) Figure 2E- It is not clear- if the memory is back what happened to it during activation of PA-Rac? Did PA-Rac1 erased the engram on the day it was activated and then it returned back spontaneously to the level before activation of PA-Rac1?

Response 1.5: In revision, we included additional data to show that upon light stimulation of PaRac1 at day 4, Rac1 is indeed activated as reflected by activation of its downstream target PAK which visualized via immunostaining of phospho-PAK (see Supplementary Fig. 6b, c). At day 7, Rac1 activation disappeared as PAK activity return to the basal level (Supplementary Fig. 6b, c). In parallel with Rac1 activity, memory is suppressed at day 4 and returned to previously level at day 7. On basis of these data, intuitively we prefer a hypothesis that memory retrieval is inhibited upon Rac1 activation instead of erased. Thus, when Rac1 activity return to basal level suppression of memory retrieval is removed.

6) "As an ideal control, manipulating the expression of α 1-chimaerin induced no significant changes in memory strength during the maintenance stage (Figures S7A-S7E) or the relevant behavioral parameters" The sentence is not clear.

Response 1.6: We revised the sentence in manuscript for clarity: "We also studied effects of manipulating another member of α -chimaerin family, α 1-chimaerin⁴. We found that manipulation of α 1-chimaerin expression had no effects on either the memory maintenance (Supplementary Fig. 8a-e, revised manuscript) or on the relevant behavioral parameters (locomotion and anxiety) (Supplementary Fig. 8f, revised manuscript)."

7) What keeps Rac1 active the whole time to maintain memory? As well as RacGAP.

Response 1.7: The activity of Rac1 is mainly regulated by Rac-specific GTPase-activating proteins (Rac-GAPs), Rac guanine nucleotide exchange factors (Rac-GEFs), and guanine-nucleotide-dissociation inhibitors (GDI)⁵. However, we have no data to determine the mechanisms associated with maintenance of learning-induced Rac1 activity. For expression of α 2-chimaerin, there is no any report (to our knowledge) for its regulatory mechanisms. We, therefore, have no idea how its learning-induced expression might be maintained.

Reviewer #2

In this manuscript, authors examined the roles of hippocampal Rac1-GTP and $\alpha 2$ -chimaerin on memory maintenance after contextual fear conditioning. First, they showed that Rac1-GTP activity is associated with the maintenance of contextual fear memory, this is consistent with their previous data using object recognition memory (Liu et al., 2016), by using global and engram-specific Rac-GTP manipulation. Authors found that $\alpha 2$ -chimaerin acts on as a crucial suppressor for the Rac1-activity for contextual fear memory by using the viral-mediated both knockdown and overexpression approaches. Consistent with behavioral data, authors also showed that induction of long-term potentiation is modulated by the artificial manipulation of $\alpha 2$ -chimaerin expression.

Although this study presents novel molecular mechanisms about how contextual memory is maintained from the aspect of dynamics for the balance between Rac1-GTP and $\alpha 2$ -chimaerin expression, I have some major and minor concerns as described below.

Major points;

1) Paper format: title and introduction are vague. Title has to be concrete, this is not review article. In introduction, Authors need to start write with previous papers about roles of Rac1-GTP and $\alpha 2$ -chimaerin on learning and memory and synaptic plasticity.

Response 2.1: As the reviewer's suggested, we revised accordingly and replaced the title "Interplay between $\alpha 2$ -chimaerin and Rac1 activity determines dynamic maintenance of long-term memory".

2) Conceptual advance: The manuscript is an extension study from previous their paper (Liu et al., 2016). Fig 1 and 2 showed re-examination with contextual fear conditioning. Fig3 and Fig4 is novel, but analysis is still weak. Author need to more focus on the role of $\alpha 2$ -chimaerin on memory maintenance as whole manuscript.

Response 2.2: As suggested, we added additional data relevant to $\alpha 2$ -chimaerin, as well as moved $\alpha 2$ -chimaerin data from supplementary to Fig. 5 in the updated manuscript. The details are presented below.

i. The effects of $\alpha 2$ -chimaerin manipulation on other behavior paradigms, including novel object recognition (Fig. 5b, revised manuscript) and three-trial CFC (Supplementary Fig. 2f, revised manuscript). We found that knocking-down $\alpha 2$ -chimaerin impairs the novel object memory paradigms (Fig. 5b, revised manuscript), while leaves the three-shock CFC-induced fear memory intact (Supplementary Fig. 2f, revised manuscript).

ii. The levels of $\alpha 2$ -chimaerin expression in three-trial CFC. We found three-trial CFC had no significant effects on $\alpha 2$ -chimaerin expression (Supplementary Fig. 2e, revised manuscript). Thus, $\alpha 2$ -chimaerin is not involved in maintenance of memory induced by three-trial CFC. This is consistent with observation that Rac1 activity is strongly inhibited in response to three-shock CFC (Supplementary Fig. 2d, revised manuscript), as well as previous reports⁶ that the forgetting of multiple-trial induced memory is mediated through neurogenesis.

iii. The mutual regulation between Rac1 activity and $\alpha 2$ -chimaerin expression. The data show that manipulation of Rac1 activity with Rac1-DN or Rac1-CA had no impact on $\alpha 2$ -chimaerin expression (Supplementary Fig. 3e, revised manuscript).

3) Cell counting issue: Overall, cell counting analysis is superficially examined. In Fig1C, is

seems all of CA1 cells shows Rac1-GTP immunoreactivity. If so, very strange, because if Rac1-GTP is essential for several type of memory, all memory will be effected. Authors need to quantify. In Fig2C and D, authors need to quantitatively count the cells which are labeled with RAC1-GTP positive in tdTomato+ and tdTomamo- cells. In Fig 3, Authors need to examine $\alpha 2$ -chimaerin positive in Ca1 engram+ and engram- cells. They should significantly overlap within same population,

Response 2.3: We agreed to the reviewer's constructive suggestion. For Fig. 1c, we performed cell counting analysis by using imaging analysis function of Zeiss software (Zen blue 2.3). We found that learning-induced Rac1-GTP immunoreactivity is about 50% in CA1 cells (Fig. 1c, revised manuscript). For Fig. 2c, d, 79% of tdTomato⁺ (c-fos⁺) engram cells were labeled with Rac1-GTP immunoreactivity and 10% of tdTomato⁻ (c-fos⁻) non-engram cells were labeled with Rac1-GTP immunoreactivity (Supplementary Fig. 5b, revised manuscript). For reviewer suggested the experiments to examine $\alpha 2$ -chimaerin positive in Ca1 engram⁺ and engram⁻ cells, there is no commercially available $\alpha 2$ -chimaerin antibody for immunohistochemistry staining. The antibody we generated is good for western blotting but failed to generate qualitative analysis for immunofluorescence (see below; Scale bar, 500 μ m).

4) In Fig 5, Authors tried to show the effect of LTP maintenance for knockdown and overexpression of $\alpha 2$ -chimaerin. However, the manipulation already caused the effect on the induction of LTP. This is inconsistent with their fear conditioning data, since it is considered protein-synthesis dependent LTP is more than 4 hours usually.

Response 2.4: We rechecked the LTP data and found that the data of fEPSP Slope (% of baseline) from the slices with $\alpha 2$ -chimaerin overexpression was not normalized. We are sorry for that. We have now renormalized all the data points to the base line average. Thus, the level of LTP induction is only slight altered by either knockdown and overexpression of $\alpha 2$ -chimaerin as shown in our revised Fig.6. Such mild difference is been reported in a number of previous publications⁷⁻¹². For instance, overexpression of calcineurin in mice shows the normal short-term memory on the novel object recognition task¹², even though induction of LTP is slightly reduced in transgenic mice⁸.

5) Authors need to show the expression level of $\alpha 2$ -chimaerin when they conduct strong CFC protocol.

Response 2.5: We detected the expression level of $\alpha 2$ -chimaerin in hippocampus following three-trial CFC. As shown in Supplementary Fig. 2e, the strong CFC had no significant effects on $\alpha 2$ -chimaerin expression. This is consistent with idea Rac1-dependent forgetting is not involved in regulation of decay of multiple-trial-induced strong memory, as Rac1 activity is not increased but strong inhibited (Supplementary Fig. 2d). (see also the response to the comment 3.2 below, because this is a related issue).

6) To generalize their finding, authors need to examine knockdown of $\alpha 2$ -chimaerin in strong CFC protocol and also object recognition memory.

Response 2.6: As suggested, we included new data of knockdown of $\alpha 2$ -chimaerin in novel object recognition and three-trial CFC (strong CFC protocol). We found that knockdown of $\alpha 2$ -chimaerin caused rapid memory decay of object recognition memory (Fig. 5c, d, revised manuscript), but the retention of fear memory after three-trial CFC remained normal (Supplementary Fig. 2f, revised manuscript).

7) While authors examined the effect of knockdown and over expression of $\alpha 2$ -chimaerin on Rac1-GTC, Authors did not show the effect of inhibition/activation of Rac1-GTP on $\alpha 2$ -chimaerin. Both $\alpha 2$ -chimaerin and Rac1-GTC may mutually regulate their activity or expression level.

Response 2.7: In accordance with the reviewer's suggestion, we explored the effect of constitutive activation or inhibition of Rac1 with Rac1-DN or Rac1-CA on $\alpha 2$ -chimaerin expression. As shown in Supplementary Fig. 3e, a viral manipulation of Rac1 activity had no impact on the expression level of $\alpha 2$ -chimaerin.

Minor points,

1) In Figure 4A to D, the authors described as AAV injection into dorsal hippocampus, but specific fluorescent signal of either tdTomato or EGFP seem to be only expressed in CA1 region without other hippocampal subregions. It should be precisely described in the article as knockdown and overexpression of $\alpha 2$ -chimaerin in "hippocampal CA1 region".

Response: We thank the reviewer for this helpful comment. We replaced "the dorsal hippocampus" with "hippocampal CA1 region" in the text.

2) Through this paper, the author frequently uses the term "middle level" memory or that memory is maintained at "middle level" to refer to the state of memory. It should be clearly described what the "middle level" means or should be rephrased with suitable word for this.

Response: We thank the reviewer for this helpful comment. We replaced "middle" with "intermediate" in revised text.

3) In line 218 of main text, (Figure S6A-S6C) should be "Figure S6A, S6B". The Figure S6C is not existed.

4) In line 228 of main text, (Figure S4D) should be "Figure S4E".

Response: We appreciate that the reviewer pointed out our mistake. We rewrote the figures in revised manuscript

5) Through discussion at the first paragraph, the authors mention to the main figures within the sentences to support their experimental evidences, but is so obscurely. It should make sure that which figure supports on your conclusive evidences.

Response: We thank the reviewer for this helpful comment. We rewrote the figures for supporting our conclusive evidences to make it clear.

- 6) In Figure 1 legend at (B), native (N) should be (“Na”).
- 7) In Figure 6, it would be better to put on “(days)” in the x-axis index.
- 8) Title for Figure S7 “after both the one and three-shock CFC” doesn’t make sense. Which graph is shown in which conditioning procedures?

Response: We appreciate that the reviewer pointed out our mistake. We replaced “(N)” with “(Na)” in legend of Fig.1 and added the title “(days)” in the x-axis index. We used the one-shock CFC procedure and rewrote the title in Supplementary Fig. 8.

Reviewer #3

In this manuscript by Lv and colleagues, the authors investigate the role of the small G-protein Rac1 in the regulation of long-term memory maintenance. The authors demonstrate using a single trial fear conditioning paradigm that memory stability is regulated by Rac1 that is upregulated following fear conditioning, with a delayed onset of expression. Using a series of pharmacological and genetic strategies, the authors demonstrate that persistent activation of Rac1 impairs long-term memory storage by facilitating forgetting. In general the results of the study are interesting and will be of broad interest, but there are a few experiments that are required to fully support the conclusions. In addition, the manuscript is written with a lot of jargon that makes it difficult to read and will be even more difficult to read by a non-expert. The insistence on the use of forgetting, as opposed to memory maintenance is particularly confusing for the reader. Forgetting is an abstract construct that is difficult to ascribe to the simple animal behavioral paradigm used here and should be avoided.¹

Response: Thanks for reviewer’s suggestion, we made efforts in an attempt to make manuscript more understandable for readers. For instance, we have replaced “middle level of memory” to “intermediate level of memory”. For the use of “forgetting” in study of memory maintenance is because the finding of the Rac1-dependent forgetting mechanism in maintaining memory. We thought such finding is significant for such forgetting-based mechanism provide base for dynamic memory maintenance. For this reason, we insisted to use forgetting in the context of memory maintenance.

Specific comments:

1) The authors use a repeated measures assay to assess memory stability. The behavioral assay confounds their analysis because the effects could be mediated by deficits in memory recall and reconsolidation. To validate that their observations are mediated by instability of the initial memory and not deficits in recall/reconsolidation in their initial tests and subsequent experiments the authors need to demonstrate using distinct cohorts of mice for each time point. One cohort for 1 hr, 1 for 1 day, 1 for 7 days, 1 for 14 days, and 1 for 36 days. If their hypothesis is correct, then the memory should be weaker in the 14 day and 36 day cohorts. This would also more directly parallel their histological and western blot analysis.

Response 3.1: In most memory stability assay, such as Fig.1a, Fig.1d, Fig. 2g-j, Fig. 4j-l, Supplementary Fig 1a, Supplementary Fig 4b, Supplementary Fig 8d, Supplementary Fig 8e and Supplementary Fig 9a, we used distinct cohorts of mice for each time point. In Fig. 1a, we have noted that separate group of mice were used to assess memory stability at various periods (1 hr, 1 day, 7 days, 14 days, or 36 days) in the legend. To make it clearer, we added a sentence in the revised text below.

“To avoid the confounding effects of extinction, separate groups of mice were tested at each time point”.

For the behavioral assay in Fig.1e, we have to collect data from one cohort, because we need to detect the effects of a Rac1 inhibitor Ehop016 on memory retention of mice with repeated Ehop016 injection for one week. We also compared the memory retention curve of group in Fig. 1a and in Fig. 1e. In fact, there is no significant difference between the memory stability from distinct cohorts and the memory stability from one cohort (see below).

2) The authors need to demonstrate that either Rac1 is not induced in the 3-trial fear conditioned mice, or that there is a persistent expression of alpha-chimaerin. This would support their claims that “forgetting” is mediated by Rac1 activation which should be not present in mice that have persistent memory.

Response 3.2: According to reviewer’s suggestion, we performed the relevant experiments and the results are as follow:

Firstly, immunoblotting showed that three-trial CFC resulted in significantly decreased Rac1 activity from 1 hour to day 1 after training (Supplementary Fig. 2d, revised manuscript), indicating inhibition of hippocampal Rac1 activity in response to three-trial CFC.

Secondly, three-trial CFC had no significant effects on α 2-chimaerin expression in hippocampus (Supplementary Fig. 2e, revised manuscript). This is consistent with the observation that knockdown of α 2-chimaerin had no effects on memory induced by three-trial CFC (Supplementary Fig. 2f). Taken together, Rac1-dependent forgetting is not involved in regulation of decay of multiple-trial-induced strong memory, as Rac1 activity is not increased but strongly inhibited.

3) The authors should demonstrate that Rac1 has the opposite effects of alpha-chimaerin on plasticity to fully support their hypothesis.

Response 3.3: In our previous publication², we have showed that compared with the ctrl slices, the increased Rac1 activity in slices expressing AAV-Rac1-CA resulted in an accelerated LTP decay, whereas the inhibition of Rac1 activity in slices expressing AAV-Rac1-DN produced a much more stable LTP that lasted over 2 hours. Therefore, Rac1 has the opposite effects of α 2-chimaerin on synaptic plasticity. This point is mentioned in the text.

References

- 1 Hashimoto, S. *et al.* Tau binding protein CAPON induces tau aggregation and neurodegeneration. *Nat Commun* **10**, 2394, doi:10.1038/s41467-019-10278-x (2019).
- 2 Liu, Y. *et al.* Hippocampal Activation of Rac1 Regulates the Forgetting of Object Recognition Memory. *Curr Biol* **26**, 2351-2357, doi:10.1016/j.cub.2016.06.056 (2016).
- 3 Liu, Y., Lv, L., Wang, L. & Zhong, Y. Social Isolation Induces Rac1-Dependent Forgetting of Social Memory. *Cell Rep* **25**, 288-295 e283, doi:10.1016/j.celrep.2018.09.033 (2018).
- 4 Beg, A. A., Sommer, J. E., Martin, J. H. & Scheiffele, P. alpha2-Chimaerin is an essential EphA4 effector in the assembly of neuronal locomotor circuits. *Neuron* **55**, 768-778, doi:10.1016/j.neuron.2007.07.036 (2007).
- 5 Yang, C. & Kazanietz, M. G. Chimaerins: GAPs that bridge diacylglycerol signalling and the small G-protein Rac. *Biochem J* **403**, 1-12, doi:10.1042/BJ20061750 (2007).
- 6 Akers, K. G. *et al.* Hippocampal neurogenesis regulates forgetting during adulthood and infancy. *Science* **344**, 598-602, doi:10.1126/science.1248903 (2014).
- 7 Jun, K. *et al.* Enhanced hippocampal CA1 LTP but normal spatial learning in inositol 1,4,5-trisphosphate 3-kinase(A)-deficient mice. *Learn Mem* **5**, 317-330 (1998).
- 8 Winder, D. G., Mansuy, I. M., Osman, M., Moallem, T. M. & Kandel, E. R. Genetic and pharmacological evidence for a novel, intermediate phase of long-term potentiation suppressed by calcineurin. *Cell* **92**, 25-37 (1998).
- 9 Okabe, S. *et al.* Hippocampal synaptic plasticity in mice overexpressing an embryonic subunit of the NMDA receptor. *J Neurosci* **18**, 4177-4188 (1998).
- 10 Shim, D. J. *et al.* Disruption of the NF-kappaB/IkappaBalpha Autoinhibitory Loop Improves Cognitive Performance and Promotes Hyperexcitability of Hippocampal Neurons. *Mol Neurodegener* **6**, 42, doi:10.1186/1750-1326-6-42 (2011).
- 11 Lynch, M. A. Long-term potentiation and memory. *Physiol Rev* **84**, 87-136, doi:10.1152/physrev.00014.2003 (2004).
- 12 Mansuy, I. M., Mayford, M., Jacob, B., Kandel, E. R. & Bach, M. E. Restricted and regulated overexpression reveals calcineurin as a key component in the transition from short-term to long-term memory. *Cell* **92**, 39-49 (1998).

Reviewers' comments:

Reviewer #1 (Remarks to the Author):

The authors responded to my comments but I have few more concerns:

- 1) Why PA-Rac1 distribution in hippocampus is not shown in figure 2f.
- 2) The effect of PA-Rac1 on p-PAK is convincing but the description of the IHC do not appear in the methods. Also the time when the p-PAK was monitored is not clear.
- 3) If PAK is activated by PA-Rac1 immediately after photoactivation by light how can post PA-Rac1 activation be affected by injection of Ehop016 (fig 2h) as PAK was already activated before by PA-Rac1? Does Ehop016 facilitate somehow the return of PA-Rac1 activated p-PAK to the basal level?
- 4) In fig 2i its not clear when is test 1 and test 2.

Reviewer #2 (Remarks to the Author):

I appreciate authors' revised manuscript. Authors extensively examined my suggestions. I think revised manuscript has been significantly improved. However, before showing a green light, I would like authors to fix following two points.

1) Wording about forgetting.

Authors used a "forgetting" many time in their manuscript without showing their definition. What is forgetting? Fail to retrieve memory or impairment of memory storage or both? To make the manuscript clear, first, I would suggest authors need to define the forgetting in introduction. Second, in result section, it is better for authors to avoid the word, because this word misleads and confuses readers' understanding. In discussion section, authors can discuss how their data support the forgetting conclusion.

In abstract, I would suggest that authors delete following sentence, since this is just duplication of previous sentences. "Stimulation of Rac1-dependent forgetting down-regulates memory whereas suppression of forgetting enhances consolidated memory"

2) 3 trials CFC:

Authors showed that 1) 3 trial CFC induced long-lasting freezing (sFig 1). 2) 3 CFC induced reduction of Rac-GTP 1 hour after the CFC, and the reduction were maintained at least for 24 hours (sFig 2). 3) Knockdown of $\alpha 2$ -chimaerin did not cause 3 trial CFC (sFig 2). From these results, authors concluded that Rac-GTP has no role for 3 trial CFC.

However, authors did not directly examine the manipulation of Rac-GTP on 3 trial CFC. For example, it is possible artificial activation of Rac-GTP after the CFC may induce the impairment of memory expression during testing, because authors clearly showed 3 trial CFC, but not shock nor context, long-lastingly reduces the expression of Rac-GTP (sFig 2). If so, it will be important evidence to show general role of Rac-GTP on memory maintenance/expression. If so, this data should go to main Fig 1.

Reviewer #3 (Remarks to the Author):

The authors have done an excellent job of addressing the main concerns raised by the reviewer in the initial submission. The manuscript now provides a stronger link between Rac1 and alpha2-chimaerin in the maintenance of long-term memory. The significant revision of the text also makes the experimental approach, the results, and interpretations more clear. The authors should be commended for their outstanding study that will have a significant impact on the field.

Reviewer #1 (Remarks to the Author):

The authors responded to my comments but I have few more concerns:

1) Why PA-Rac1 distribution in hippocampus is not shown in figure 2f.

Response 1.1: The Fig. 2f shown of tdTomato is in fact PaRac1 distribution, but the labelling is misleading. To make it clearer, we replaced “tdTomato” with “PaRac1-tdTomato” in revised Fig. 2f. Moreover, we added the data relevant to Control AAV (Control-tdTomato) distribution in hippocampus (Fig. 2f, revised manuscript).

2) The effect of PA-Rac1 on p-PAK is convincing but the description of the IHC do not appear in the methods. Also the time when the p-PAK was monitored is not clear.

Response 1.2: We revised the description of the IHC for p-PAK in methods and we also added the experimental schedule for the time when the p-PAK was monitored (see Supplementary Fig. 6b, c and Methods).

3) If PAK is activated by PA-Rac1 immediately after photoactivation by light how can post PA-Rac1 activation be affected by injection of Ehop016 (fig 2h) as PAK was already activated before by PA-Rac1? Does Ehop016 facilitate somehow the return of PA-Rac1 activated p-PAK to the basal level?

Response 1.3: To address the reviewer’s concern, we explored the effect of Ehop016 on the level of PaRac1 activated p-PAK. As shown in Supplementary Fig. 6b-d, Ehop016 facilitates the return of activated p-PAK to the basal level, in comparison to mice without Ehop016 injection.

4) In fig 2i its not clear when is test 1 and test 2.

Response 1.4: We revised the Figure 2 legend in manuscript for clarity (Fig. 2i, revised manuscript).

Reviewer #2 (Remarks to the Author):

I appreciate authors' revised manuscript. Authors extensively examined my suggestions. I think revised manuscript has been significantly improved. However, before showing a green light, I would like authors to fix following two points.

1) Wording about forgetting.

Authors used a "forgetting" many time in their manuscript without showing their definition. What is forgetting? Fail to retrieve memory or impairment of memory storage or both? To make the manuscript clear, first, I would suggest authors need to define the forgetting in introduction. Second, in result section, it is better for authors to avoid the word, because this word misleads and confuses readers' understanding. In discussion section, authors can discuss how their data support the forgetting conclusion.

In abstract, I would suggest that authors delete following sentence, since this is just duplication of previous sentences. "Stimulation of Rac1-dependent forgetting down-regulates memory whereas suppression of forgetting enhances consolidated memory"

Response 2.1: As the reviewer's suggested, we revised in abstract, introduction, result and discussion accordingly (revised manuscript). Firstly, we included the sentence for the definition of the forgetting in the introduction section. Secondly, we removed the word of forgetting in most cases in the result section. Finally, we deleted the sentence in the abstract as suggested.

2) 3 trials CFC:

Authors showed that 1) 3 trial CFC induced long-lasting freezing (sFig 1). 2) 3 CFC induced reduction of Rac-GTP 1 hour after the CFC, and the reduction were maintained at least for 24 hours (sFig 2). 3) Knockdown of $\alpha 2$ -chimaerin did not cause 3 trial CFC (sFig 2). From these results, authors concluded that Rac-GTP has no role for 3 trial CFC.

However, authors did not directly examine the manipulation of Rac-GTP on 3 trial CFC. For example, it is possible artificial activation of Rac-GTP after the CFC may induce the impairment of memory expression during testing, because authors clearly showed 3 trial CFC, but not shock nor context, long-lastingly reduces the expression of Rac-GTP (sFig 2). If so, it will be important evidence to show general role of Rac-GTP on memory maintenance/expression. If so, this data should go to main Fig 1.

Response 2.2: In the old discussion section, we indeed suggest that Rac-GTP may not contribute to 3-trial CFC. Our preliminary exploration of 3-trial CFC indicated that the underlying mechanisms appeared to be more complex than we thought before and it alone is becoming an independent project. Therefore, we thought it is better to remove the suggestion in the discussion section and such deletion has no impact on the significance of manuscript since the major scope of this manuscript is confined to one-trial CFC.

REVIEWERS' COMMENTS:

Reviewer #1 (Remarks to the Author):

I find that my concerns have been addressed appropriately in the revised manuscript.

Reviewer #2 (Remarks to the Author):

The authors have examined an excellent job of addressing the my concerns.